# CausalDynamics: A large-scale benchmark for structural discovery of dynamical causal models

**Benjamin Herdeanu**[*]
kausable GmbH
benjamin@kausable.ai

**Juan Nathaniel**[*]
Columbia University
jn2808@columbia.edu

**Carla Roesch**[*]
Columbia University
cmr2293@columbia.edu

**Jatan Buch**
Aeolus Labs
jatan@aeolus.earth

**Gregor Ramien**
kausable GmbH
gregor@kausable.ai

**Johannes Haux**
kausable GmbH
johannes@kausable.ai

**Pierre Gentine**
Columbia University
pg2328@columbia.edu

## Abstract

Causal discovery for dynamical systems poses a major challenge in fields where active interventions are infeasible. Most methods used to investigate these systems and their associated benchmarks are tailored to deterministic, low-dimensional and weakly nonlinear time-series data. To address these limitations, we present *CausalDynamics*, a large-scale benchmark and extensible data generation framework to advance the structural discovery of dynamical causal models. Our benchmark consists of true causal graphs derived from thousands of both linearly and nonlinearly coupled ordinary and stochastic differential equations as well as two idealized climate models. We perform a comprehensive evaluation of state-of-the-art causal discovery algorithms for graph reconstruction on systems with noisy, confounded, and lagged dynamics. *CausalDynamics* consists of a plug-and-play, build-your-own coupling workflow that enables the construction of a hierarchy of physical systems. We anticipate that our framework will facilitate the development of robust causal discovery algorithms that are broadly applicable across domains while addressing their unique challenges. We provide a user-friendly implementation and documentation on `https://kausable.github.io/CausalDynamics`.

## 1 Introduction

Understanding causal mechanisms in nonlinear dynamical systems is crucial across many fields. However, direct interventions are often impractical or impossible. An alternate strategy for encoding causal information is through detailed physical modeling, such as Earth System models, which relies on ad-hoc parameterizations and can be computationally expensive. In contrast, data-driven causal discovery frameworks have emerged as a promising avenue to infer cause–effect relationships directly from time series observations, obviating the need for interventions or simulations. Originating from Wright's path analysis in the 1930s [1] and later formalized by Wiener [2] and Granger [3] in the mid-20$^{\text{th}}$ century, causal inference now includes deep learning (DL)-based methods [4, 5], designed to infer complex, nonlinear, and stochastic causal structures.

---

[*]Equal contribution

39th Conference on Neural Information Processing Systems (NeurIPS 2025) Track on Datasets and Benchmarks.

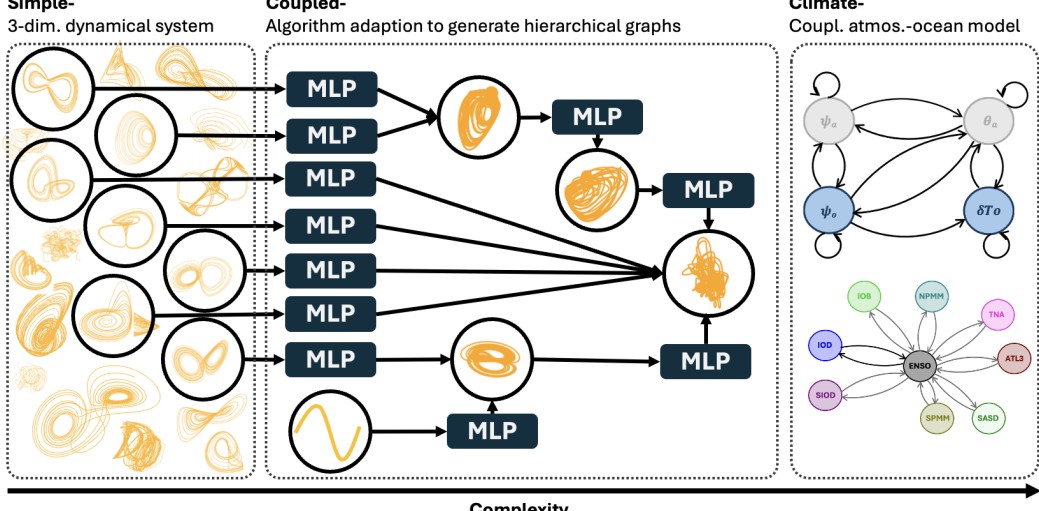

Figure 1: Illustration of the tiered framework in **CausalDynamics** consisting of a plug-and-play, build-your-own coupling workflow that enables the construction of a hierarchy of physical systems with common causal challenges, such as unobserved confounders, time-lags, and noisy time series.

Despite rapid methodological advances, there exists no standardized benchmark for evaluating causal discovery in highly nonlinear, dynamical settings [6, 7, 8], which is critical to understand and predict the behavior of physical systems [9, 10, 11, 12, 13, 14]. Most benchmarks are built on synthetic data generated by static causal graphs (e.g., [15]) or auto-regressive models (e.g., [16, 17]) for domain-specific applications [18, 19]. Limited examples from the real-world are available but lack a fully resolved causal ground truth [20, 21, 22]. As a result, most methods are validated on toy systems or on real-world data failing to capture continuous state-space developments, complex feedback loops, stochasticity, and regime shifts, making it impossible to isolate algorithmic limitations (e.g., varsortability [23, 24]) from dataset characteristics.

We posit that advancing the development of new causal discovery methods capable of capturing high dimensional, nonlinear, time-lagged, sparse, and noisy structures with unobserved confounding, often found in real-world physical systems [25, 26, 27], requires a novel benchmark. In fact, examples from various fields demonstrate how such benchmarks also play a vital role in unlocking methodological innovation for data-driven methods and boosting scientific progress. A prominent example includes CASP13, a protein structure modeling benchmark that enabled the development of AlphaFold, a co-awardee of the 2024 Nobel Prize in Chemistry [28, 29, 30]. Other benchmarks include ImageNet [31] which highlighted the computer vision capabilities of AlexNet [32] or NWPU-RESISC45 which first demonstrated the superior skill of deep learning approaches for remote-sensing classification [33].

In this paper, we present **CausalDynamics**, a tiered benchmark and extensible data generation framework, as shown in Figure 1, to advance the structural discovery of dynamical causal models. The simple tier contains true causal graphs for hundreds of three-dimensional chaotic dynamical systems. In the second tier, we adapt an existing algorithm [34] for graph generation to couple deterministic and stochastic dynamical systems with periodic functions for constructing thousands of complex graph structures. This framework allows us to address the set of challenges often found in real-world systems as highlighted previously. The final tier contains the true causal graphs for two idealized atmosphere-ocean models including multiple coupling experiments of different modes of variability.

Simultaneously generating pseudo-real or idealized data has a two-fold advantage: (i) We can benchmark causal discovery methods with known causal graphs and well-designed challenges, and (ii) once calibrated, methods can be extended to realistic applications. This way, we ensure a comprehensive workflow for developing trustworthy, novel causal discovery methods. **CausalDynamics** provides that framework and builds a robust foundation for testing any novel algorithm's applicability to real-world scenarios.

Our contributions include:

- **CausalDynamics**, the largest benchmark for causal discovery methods for stochastic chaotic systems containing over 14000 graphs and over 50 million preprocessed samples.
- A novel graph generating algorithm that allows users to easily extend the complexity of the synthetic dataset as more powerful causal discovery methods are developed.
- An all-in-one benchmark for evaluating causal discovery methods such that any new algorithm can be reliably evaluated on synthetic data with known graph structure before being tested on a real-world example within the same framework.
- Evaluation of the performance of several state-of-the-art (SOTA) DL-based and non-DL-based causal discovery algorithms for graph reconstruction on our benchmark.

A quickstart guide to **CausalDynamics** can be found in Appendix A and at `https://kausable.github.io/CausalDynamics/notebooks/quickstart.html`.

## 2 Theoretical background

In this section, we will give a brief background on the connection between dynamical systems and structural dynamical causal models [35] to introduce the necessary notation.

### 2.1 Structural dynamical causal model

**Dynamical system.** For each time $t \in \mathbb{R}_{\geq 0}$, we characterize a dynamical system with an associated state $x(t) \in \mathbb{R}^N$ for $N \in \mathbb{N}$. In general, the description of the system dynamics are given through differential equations of the form:

$$\frac{dx}{dt} = f(t, x) + \delta \frac{dW_t}{dt} \tag{1}$$

where $f$ is a function, the solution $x(t)$ depends on the initial condition $x(t_0) = x_0$ at time $t_0$, and $\delta$ is the noise amplitude of the Brownian process $W_t$ [36, 37, 38]. When $\delta = 0$, Equation 1 becomes *ordinary differential equations* (ODEs) whereas $\delta > 0$ yields *stochastic differential equations* (SDEs).

**Structural causal models.** We describe causal mechanisms through structural causal models (SCMs) such that a system of $d$ random variables $\boldsymbol{x} = \{x_1, \ldots, x_d\}$ is expressed as an arbitrary function $f^k$ of its direct parents (causes) $\boldsymbol{x}_{\mathrm{PA}_k}$ and an exogenous distribution of noise $\epsilon^k$ [35]:

$$x_k := f^k(\boldsymbol{x}_{\mathrm{PA}_k}, \epsilon^k), \text{ for } k = 1, \ldots, d \tag{2}$$

For dynamical systems, we can extend Equation 2 with Equation 1 for a collection of $d$ differential equations to define *structural dynamical causal models* (SDCM) [35, 39, 40]:

$$\frac{\mathrm{d}}{\mathrm{d}t} x_{k,t} := f^k(\boldsymbol{x}_{\mathrm{PA}_k, t}, \delta), \text{ with } x_{k,0} = x_k(0) \tag{3}$$

where $k \in \{1, \ldots, d\}$.

**Causal graph.** The structural assignment of the SCM induces a *directed acyclic graph* (DAG) $\mathcal{G} = (\mathcal{V}, \mathcal{E})$ over the variables $x_k$. $\mathcal{G}$ includes nodes $v_k \in \mathcal{V}$ for every $x_k \in \boldsymbol{x}$ and directed edges $(k, i) \in \mathcal{E}$ if $x_k \in \boldsymbol{x}_{\mathrm{PA}_i}$ [23]. Edges are represented in a squared *adjacency matrix* $\mathcal{A} \in \mathbb{R}^{k \times i}$ with each entry $a_{ki} \in \mathcal{A} : a_{ki} = 1$ if $x_i$ is causally impacted by $x_k$, else $a_{ki} = 0$. A corresponding graphical representation of the DAG is shown in Figure 2.

### 2.2 Causal challenges

We include a set of causal challenges that are common in real-world physical systems to maximize the applicability of our benchmark. The challenges include: (i) *Noise* can obscure statistical dependencies or identify spurious links [41]; (ii) *Hidden confounders*, i.e., unobserved variables that are a common cause of at least two other unrelated variables, induce correlations that algorithms may misinterpret as direct causal links [42, 43, 25]; (iii) Delays between cause and effect, i.e., *time-lags*, lead to causal

$$\dot{x} = -y - z$$
$$\dot{y} = x + ay \quad \Rightarrow \quad \mathcal{A} = \begin{bmatrix} 0 & 1 & 1 \\ 1 & 1 & 0 \\ 1 & 0 & 1 \end{bmatrix} \quad \Leftrightarrow$$
$$\dot{z} = b + zx - cz$$

Figure 2: Illustration of the Rössler Oscillator ODEs (left), the associated adjacency matrix $\mathcal{A}$ (center) and the corresponding causal graph (right).

effects at multiple time scales, which obscure conditional independence test and result in spurious links [44]. Lastly, varsortability is an artifact in synthetically generated data from SCMs describing the increase in a variable's variance along its topological order [45, 46]. Therefore, we also implement the option to (iv) *standardize* [23, 24].

# 3 Related work

In this section, we review existing benchmarks for causal discovery, highlighting their shortcomings for dynamical systems, and present an overview of existing causal discovery methods.

## 3.1 Benchmarks for causal discovery

**Real-world.** Real-world datasets for causal discovery remain rare because true causal graphs are difficult to obtain. A handful of benchmarks provides *time series* data, e.g. human-motion capture (MoCap) [5], S&P 100 stock prices [22], climate variability and teleconnections [47], river discharge (CausalRivers) [20], and ecological observations [48]. However, most observed collections focus on *static*, domain-specific graphs, such as large-scale single-cell RNA perturbations (CausalBench) [18, 19], the Old Faithful geyser eruptions [49] and immune-cell protein networks [50, 51]. Bivariate settings like the Tübingen Cause Effect Pairs [38] further complement these resources.

**Synthetic.** Due to the domain specifications, limited availability and often low dimensionality of real-world datasets, synthetic data plays an important role in understanding complex systems in general [52, 53, 54, 55] and benchmarking causal discovery methods [6] specifically. In *static* settings, SCMs or structural vector autoregressive (SVAR) models sample causal coefficients and noise to generate graphs [24] often following multivariate linear or nonlinear functions [49, 56, 57], or physical laws [58]. More realistic pipelines derive the causal graph to fit coefficients from observational data (e.g. causalAssembly [15], SynTReN [59]), which limits their application to the domain and the properties of the underlying observed dataset. For *time series*, idealized dynamical systems, e.g., Lorenz or Rössler Oscillators [60], underpin simpler benchmarks [5, 61, 62, 63], while domain-informed models include fMRI networks (Netsim) [64], gene regulatory networks (DREAM3 & 4) [65, 66] or financial data (FINANCE) [67, 68]. A collection of climate and weather benchmarks can be found on CauseMe (https://causeme.uv.es/) [21, 69] with pseudo-realistic climate data generated using the SAVAR model [17]. Going beyond single-domain applications, novel approaches propose pipelines to flexibly generate time series datasets for a range of properties and natural systems [70] from any observational dataset using DL (e.g., CausalTime [16]).

Time series data is critical for benchmarking SDCM discovery algorithms. However, as outlined above, relevant benchmarks contain only domain specific datasets consisting of a small number of graphs [64, 17, 16] and weakly nonlinear systems [20]. Further, even though datasets might contain thousands of samples [18], the underlying graphs are but a handful. Although observation-based approaches generate pseudo-real data for a range of domains, they lack a reliable ground truth validation [16, 70].

## 3.2 Causal discovery methods

In the following section, we provide a summary of the different causal discovery approaches which can be categorized into five classes. A detailed review of existing methods can be found in [4].

(i) *Granger causality* (GC) [3] is one of the oldest concepts in causal inference [71]. GC tests whether a given effect is optimally forecast by its causes under the assumptions of no unobserved confounders. GC often fails for nonlinear dynamics [48], leading to deep-learning variants: Neural GC (NGC) [5] employs regularized multilayer perceptrons (cMLP) or long short-term memory (cLSTM) to learn nonlinear autoregressive links, while CUTS+ uses a coarse-to-fine pipeline with a message-passing graph neural network to recover causal graphs from high-dimensional, irregular time series [72]. Another notable time-series extension is Temporal Causal Discovery Framework (TCDF) [68], which leverages convolutional neural networks with attention mechanisms to identify lagged causal links directly from multivariate sequences.

(ii) *Constraint-based* methods infer causal structure by enforcing the statistical constraints implied by conditional independencies in the data (e.g., PC [41]). PCMCI+ (Peter Clark Momentary Conditional Independence) [44] adapts PC to multivariate time series by preselecting candidate parents and performing Momentary Conditional Independence (MCI) tests over all time-lags to infer contemporaneous and lagged edges. F-PCMCI boosts scalability by prefiltering parents via transfer-entropy before MCI testing [73]. More recently, permutation-based methods such as Greedy Relaxations of the Sparsest Permutation (GRaSP) [74] have been developed, which search over permutations of variables and iteratively prune spurious edges to identify sparse causal graphs.

(iii) *Noise-based* causal discovery methods exploit the fact that in a correctly specified SCM, the noise term is statistically independent of its inputs only in the true causal direction. For example, LiNGAM [75] and its temporal extension VARLiNGAM [76] assume a linear model with non-Gaussian noise and recover a unique DAG and lagged links by analysis the residuals, while additive noise models (ANMs) (e.g., [49]) generalize this to nonlinear relations by choosing the direction where residuals remain independent of their input. The recent Repetitive Causal Discovery (RCD) algorithm [77] extends the LiNGAM family by incorporating constrained functional causal models with regularization, enabling more robust identification in high-dimensional or noisy settings.

(iv) *Score-based* learning algorithms infer causal relationships between variables by evaluating and ranking potential causal graphs based on a score function such as least-squares error. For instance, DYNOTEARS [22] adopts a score-based SVAR formulation with a penalized least-squares loss and a differentiable acyclicity constraint to jointly recover instantaneous and lagged weights.

(v) *Topology-based* approaches are based on Takens' theorem [78] to reconstruct the attractor of a dynamical system from delay-embedded time series data. TSCI (Tangent Space Causal Inference) [79] then applies Convergent Cross Mapping [48] within each manifold's tangent space to test for causal influence by assessing how well the state-space of one variable predicts another.

## 4  *CausalDynamics*

In the following, we introduce **CausalDynamics**, a large-scale benchmark and extensible data generation framework as illustrated in Figure 1. We produce three tiers of datasets, each introducing a progressively greater level of complexity as outlined in the rest of the section. Our benchmark consists of true causal graphs derived from thousands of coupled ordinary and stochastic differential equations (tiers 1 & 2) as well as two idealized climate models (tier 3). The code is available at `https://github.com/kausable/CausalDynamics` and the dataset can be downloaded from `https://huggingface.co/datasets/kausable/CausalDynamics`. Full documentation can be found at `https://kausable.github.io/CausalDynamics`. A sample will also be available on CauseMe [44].

### 4.1  Tier 1 – Simple causal models

In the *simple* complexity tier, we select 59 three-dimensional systems from the `dysts` benchmark [60, 80] and derive their SDCMs as adjacency matrices. For each system, we simulate trajectories with 5 random initial conditions over 1000 time steps.

We perform two experiments in this tier: (i) *unobserved confounding* by excluding the respective time series during evaluation ($x$ in the system in Figure 2), and (ii) we evaluate for *varying Langevin noise amplitude* ($\delta$) exploiting the `dysts` support for both deterministic and stochastic integration schemes.

**Algorithm 1** Growing Network with Redirection (GNR) model from [34]

---

**Require:** $n \in \mathbb{N}, r \in [0, 1]$
1:  $\mathcal{A} \leftarrow \text{zeros}(n \times n); K \leftarrow \text{zeros}(n); v_a \leftarrow \text{zeros}(n)$  ▷ adj. matrix; attachm. kernel; ancestors
2:  **if** $n = 1$ **then**                                                      ▷ Case single node
3:     $\mathcal{A} \leftarrow [[0]]; K \leftarrow [0]; v_a \leftarrow [0]$
4:     **return** $\mathcal{A}$
5:  **end if**
6:  $K[0] \leftarrow 1; v_a[0] \leftarrow 0$                                 ▷ Initialize attachment kernel and ancestors
7:  **for** $i \leftarrow 1$ to $n - 1$ **do**
8:     $K^{\text{prob}} \leftarrow K[0 : i] / \sum(K[0 : i])$                     ▷ Normalize to get probabilities
9:     $m \leftarrow \text{Multinomial}(K^{\text{prob}}, 1)$                     ▷ Sample node based on probabilities
10:    **if** $\text{random}(n - 1) < r = \text{True}$ **then**
11:       $m \leftarrow v_a[m]$                                              ▷ Redirect to ancestor node
12:    **end if**
13:    $\mathcal{A}[i, m] \leftarrow 1$                                        ▷ Add edge in graph
14:    $K[i] \leftarrow K[i] + 1; K[m] \leftarrow K[m] + 1$       ▷ Update source & target attachment kernels
15:    $v_a[i] \leftarrow m$                                                  ▷ Track ancestor
16: **end for**
17: **return** $\mathcal{A}$

---

## 4.2 Tier 2 – Hierarchically coupled causal models

In the second complexity tier, we introduce hierarchically *coupled* causal models, which move beyond isolated dynamical systems and simulate causal interdependence between multiple driving processes. For this, we adapt an algorithm first proposed by [34] and extend it to conduct flexible experiments on pseudo-real causal challenges.

**Sampling.** Coupled models are sampled using a Growing Network with Redirection (GNR) model following [34], which we refer to as *scale-free* DAGs due to their underlying preferential attachment nature [81] with redirection mechanisms. The redirection probability $r$ controls the balance between preferential attachment and ancestor-based connections. We outline the GNR model for graph generation in Algorithm 1 and visualized in Figure 3.

**Causal units.** Each node $v_k \in \mathcal{V}$ represents a causal unit consisting of $d$ time series $x_{v_k}(t) \in \mathbb{R}^d$ for $k \in n$, where $d = \{1, 3\}$.

Root nodes, i.e., nodes without incoming edges, are initialized using one of the following drivers (Figure 3a):

- *Dynamical drivers:* Chaotic systems sampled from the `dysts` package (tier 1), such as Lorenz or Rössler ($d = 3$).

- *Periodic drivers:* Sinusoidal functions of the form $x_{v_k}(t) = A \sin(\omega t + \phi)$, capturing seasonal or oscillatory influences ($d = 1$).

- *Linear drivers:* Linear functions $x_{v_k}(t) = mt + b$ modeling continuous temporal changes such as rising global mean temperatures in first order approximation.

**Edges.** Non-root nodes integrate the transformed signals received from their parent nodes via the information carried by the edges [82, 83]. Each edge $(k, i) \in \mathcal{E}$ is realized as an MLP [84] with optional activations $\phi_{(k,i)}$:

$$f_{(k,i)}(x_k(t)) = \phi_{(k,i)} \left( W_{(k,i)} x_k(t) + b_{(k,i)} \right),$$
$$W_{(k,i)} \sim \mathcal{N}(0, 1), \ b_{(k,i)} \sim \mathcal{N}(0, 1), \tag{4}$$
$$\phi_{(k,i)} \sim \text{Uniform} \left( \{\text{identity}, \sin, \text{sigmoid}, \tanh, \text{ReLU}\} \right)$$

where $W_{(k,i)} \in \mathbb{R}^{d \times d}$ and $\phi_{(k,i)} = \text{identity}$ in case of no activations. Weights are sparsified with a dropout probability $p_{\text{zero}}$. The value of node $v_k$ at time $t$ is computed by aggregating its incoming signals:

$$x_{v_k}(t) = \sum_{k \in \text{pa}_i} f_{(k,i)}(x_k(t)). \tag{5}$$

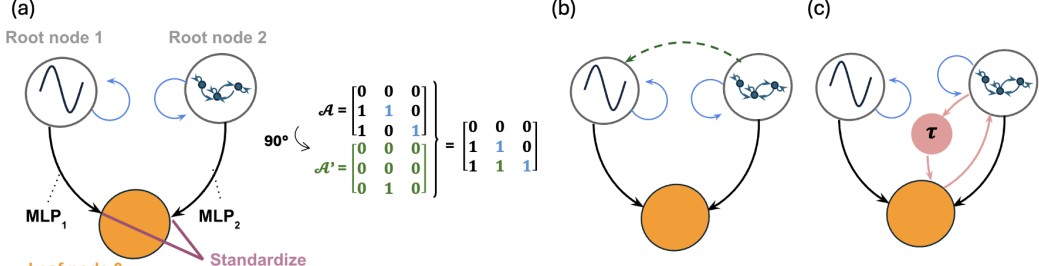

Figure 3: (a) Hierarchically coupled graphs are sample as scale-free DAGs, where root nodes can either be driven by period functions (root node 1) or dynamical systems (root node 2). Information is passed to the leaf nodes (leaf node 0) through MLPs. To correct for varsortability node values can optionally be standardized. To create a diverse set of challenges we introduce (b) unobserved confounders (green dotted lines), or (c) time-lag $\tau$ (red node). The type of nonlinearity can also be varied by prescribing or randomly assigning different edge-level activation functions. Assigned drivers are plotted by solid black edges. Note that each node is $\in \mathbb{R}^d$ and each edge is a function $f : \mathbb{R}^d \rightarrow \mathbb{R}^d$.

Following this flexible coupling we can generate the following experiments:

- *Confounder:* By sampling two adjacency matrices, i.e. two sets of edges, for a set of nodes, rotating the off-diagonal entries in the second matrix by 90 degrees, and then combining the two, we can introduce confounders as visualized in Figure 3b. This method also ensures that the confounded graph is a scale-free DAG.

- *Time-lag:* To model delayed effects, we introduce time-lagged edges as shown in Figure 3c. For a fixed lag $\tau$, selected edges introduce temporal delay:

$$x_{v_k}(t) = f_{(k,i)}(x_k(t - \tau)), \tag{6}$$

  breaking the acyclic constraint over time and introducing temporally cyclic graphs. The delay $\tau$ is constant per graph instance and such edges are sampled with probability $p_t$.

- *Standardized:* Similar to [23, 24] we standardize node values to remove scaling artifacts over the time dimension:

$$\tilde{x}_{v_k}(t) = \frac{x_{v_k}(t) - \mu_{v_k}}{\sigma_{v_k}}, \tag{7}$$

  where $\mu_{v_k}$ and $\sigma_{v_k}$ are the mean and standard deviation over time.

More details on data generation for coupled systems, including pseudocode, graphical illustrations, and a detailed sketch of the procedure of standardization are included in Appendix B.

### 4.3 Tier 3 – Pseudo-real physical systems

In the highest tier, we generate pseudo-real *climate* data of coupled atmosphere–ocean dynamics. The Earth's climate is a high-dimensional, non-equilibrium, chaotic, complex system and its predictability remains an active area of research [85, 86, 87, 88].

In a low-dimensional setting, we model the El Niño–Southern Oscillation (ENSO) as implemented in the XRO package [89], which merges the Hasselmann stochastic framework [90] and recharge oscillator dynamics [91]. As XRO initializes with observed sea surface temperatures and thermocline depth, it reproduces key observational ENSO statistics [92]. The package also links multiple ocean-basin modes, allowing for adjustable coupling strengths to simulate tightly coupled or largely independent behavior. To simulate higher dimensional atmosphere-ocean dynamics, we utilize the qgs package of a simplified quasi-geostrophic two-layer model resolving barotropic and baroclinic interactions derived from the Modular Arbitrary-Order Ocean-Atmosphere Model (MAOOAM) [93].

Between XRO's observations-based stochastic, oscillatory dynamics and qgs's high dimensional coupling, our benchmark includes datasets that closely mimic both observed dynamical systems and operational climate model outputs. Appendix C contains more details of the implemented models.

### 4.4 Dataset summary

We provide a summary of the preprocessed dataset in Table 1 and Figure 13. Unless otherwise stated, each graph constitutes 5 randomly initialized trajectories with 1000 time steps. In total, we generated 585 simple, 14096 coupled, and 12 climate graphs, for a total of 14693 graphs. More details on the selected parameters can be found in Appendix D.

Table 1: Complexity tiers in *CausalDynamics*.

| Tier | Model | Challenges | # Graphs |
|------|-------|-----------|----------|
| Simple | ODE/SDE | Confounder | 585 |
| Coupled | Coupled ODE/SDE ($N=\{3, 5, 10\}$) | Confounder, time-lag, standardized, forcing | 14096 |
| Climate | MAOOAM + ENSO models | High dimensionality | 12 |

### 4.5 Evaluation metrics

Our benchmark is concerned with reconstructing the true causal graph of the underlying dynamical systems. To evaluate the similarity of the graphs inferred by the baselines and the true causal graph we estimate Area Under the Receiver Operating Characteristic (AUROC) [94, 95], Area Under the Precision Recall Curve (AUPRC) [96], and Structural Hamming Distance (SHD) [97, 98] scores by comparing the true and predicted adjacency matrices. Our full evaluation workflow including details on baseline methods is provided in Appendix E.

### 4.6 Limitations

We only processed data that is based on a fixed set of parameters (e.g., time-lag, noise level) and 3-dimensional systems. This is only a choice reflecting common challenges, as our plug-and-play framework provides options for users to scale the generated data to unrestricted complexity.

## 5 Experiments

To showcase how SOTA algorithms handle the various challenges of our benchmark, we evaluate the approaches described in Section 3.2, reporting a selection of the results in Tables 2-3 and the full results online at `https://huggingface.co/datasets/kausable/CausalDynamics`. The default setup for the simple case refers to ODE systems ($\delta = 0$) with no unobserved confounder. In the coupled systems tier, the default setup refers to dynamics with $n = 10$ coupled ODEs ($\delta = 0$), no confounder, no time-lag ($\tau = 0$), and no standardization. We performed an initial hyperparameter search for most of the baseline algorithms and provide the results in Section E.1.

We find methods that do not require much fine tuning, such as the non-DL-based algorithms, perform better as illustrated in Tables 2-3 and Figure 4, and observed by [20]. This is striking for the coupled

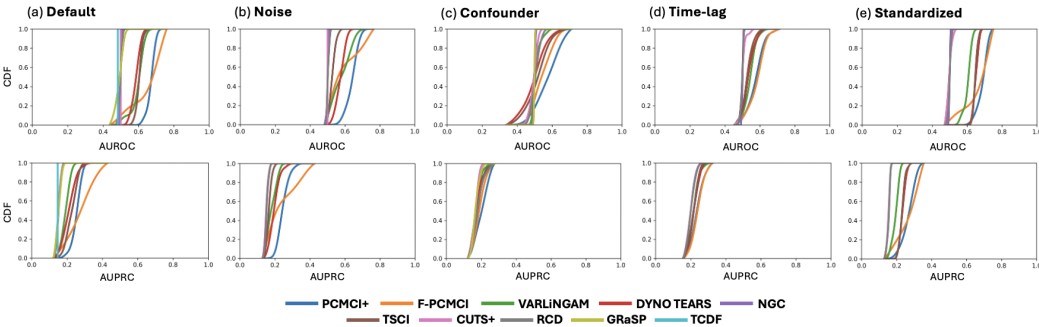

Figure 4: CDF of AUROC (top) and AUPRC (bottom) for mean values reported in Table 2 for different baselines across coupled system experiments. Models that perform better yield a low area under the curve for both AUROC and AUPRC.

Table 2: Baseline AUROC (↑) / AUPRC (↑) scores across different experiments in the hierarchy of increasingly complex dynamical systems. The scores are averaged over all generated graphs within a specific experiment. The experiments reported here consider one causal problem at a time, while keeping other factors at the default setting. The full results, including all possible experimental permutations, are found in our data repository.

| | PCMCI+ | F-PCMCI | VARLiNGAM | DYNOTEARS | NGC | TSCI | CUTS+ | RCD | GRaSP | TCDF |
|---|---|---|---|---|---|---|---|---|---|---|
| **Experiments** | | | | | **Simple** | | | | | |
| Default | **.52 / .71** | .51 / .70 | .50 / .69 | .43 / .67 | .50 / .69 | .46 / .68 | .50 / .69 | .50 / .69 | .52 / .71 | .51 / .70 |
| Noise | .50 / .69 | .52 / .70 | **.53 / .70** | .48 / .68 | .50 / .68 | .49 / .68 | .50 / .68 | .50 / .68 | .49 / .68 | .50 / .68 |
| Confounder | .49 / .59 | .50 / .59 | .48 / .58 | .52 / .64 | .50 / .58 | .53 / .66 | .50 / .58 | .50 / .58 | **.55 / .64** | .50 / .58 |
| | | | | | **Coupled** | | | | | |
| Default | .67 / .25 | **.67 / .27** | .60 / .19 | .59 / .21 | .50 / .15 | .60 / .23 | .50 / .15 | .50 / .15 | .49 / .15 | .48 / .15 |
| Noise | **.64 / .25** | .57 / .21 | .57 / .18 | .57 / .20 | .50 / .15 | .53 / .17 | .50 / .15 | .50 / .15 | .50 / .50 | .50 / .50 |
| Confounder | **.58 / .20** | .55 / .19 | .51 / .17 | .49 / .17 | .50 / .16 | .51 / .18 | .49 / .16 | .51 / .18 | .50 / .17 | .50 / .50 |
| Time-lag | .58 / .24 | **.59 / .24** | .54 / .22 | .53 / .22 | .50 / .20 | .53 / .21 | .50 / .20 | .50 / .20 | .50 / .50 | .50 / .50 |
| Standardize | **.69 / .27** | .68 / .28 | .60 / .19 | .65 / .23 | .50 / .15 | .65 / .23 | .50 / .15 | .50 / .16 | .50 / .50 | .50 / .50 |
| | | | | | **Climate** | | | | | |
| MAOOAM | **.69 / .88** | .50 / .81 | .50 / .81 | .64 / .86 | .50 / .81 | .58 / .84 | .50 / .81 | .50 / .81 | .48 / .81 | .50 / .81 |
| ENSO | .57 / .70 | **.57 / .70** | .56 / .69 | .55 / .69 | .50 / .67 | .50 / .67 | .50 / .67 | .50 / .67 | .50 / .50 | .50 / .50 |

atmosphere-ocean model, where each node represents a 2-dimensional spatial field. Methods that only consider 1-dimensional time-series, perform best, compared to DL-based methods, which claim to have a notion of space. We also find some evidence that topology-based methods, such as TSCI, perform better than purely DL-based approaches, for confounded and higher dimensional systems.

Overall, existing algorithms show shortcomings when confronted with coupled dynamical and physical systems as shown in Figure 5 for a random single graph realization. Across methods we find that autocorrelation is inferred where none exists. The baselines also appear conservative, predicting dense adjacency in the presence of nonstationary dynamics. In the climate case, methods perform well but do not recover the full coupling between ocean basins. Nevertheless, scores are the highest for the climate examples, possibly due to the noise in the generated data which might benefit some methods in recovering the underlying graph structure. Surprisingly in Figure 4, we observe minimal improvement in the standardized case, especially for methods like VARLiNGAM that exploit topological ordering. We performed additional ablation studies that consider the effect of time subsampling, partial observability, and varying edge-level activation function in Appendix F.

Finally, we note that for sparse graphs, AUROC can be deceptively high because the false-positive rate remains low due to a higher number of true-negatives. Thus, for coupled systems AUPRC scores might be more representative of the true performance due to the penalization of false positives. Other metrics like SHD [97, 98] or Structural Intervention Distance [99], suffer from different limitations

Table 3: Baseline SHD (↓) score across different experiments in the hierarchy of increasingly complex dynamical systems. The scores are averaged over all generated graphs within a specific experiment. The experiments reported here consider one causal problem at a time, while keeping other factors at the default setting. The full results, including all possible experimental permutations, are found in our data repository.

| | PCMCI+ | F-PCMCI | VARLiNGAM | DYNOTEARS | NGC | TSCI | CUTS+ | RCD | GRaSP | TCDF |
|---|---|---|---|---|---|---|---|---|---|---|
| **Experiments** | | | | | **Simple** | | | | | |
| Default | 41.04 | 35.30 | 35.96 | 52.37 | **28.91** | 52.50 | 48.11 | 61.85 | 59.04 | 59.46 |
| Noise | 183.90 | **149.60** | 248.90 | 180.95 | 842.55 | 173.50 | 150.50 | 155.65 | 842.55 | 842.55 |
| Confounder | 23.02 | 21.07 | 22.04 | 21.74 | **19.96** | 22.02 | 24.04 | 26.74 | 27.09 | 26.61 |
| | | | | | **Coupled** | | | | | |
| Default | 224.80 | 192.90 | 311.45 | 181.50 | 840.95 | 174.70 | **152.00** | 157.05 | 215.94 | 252.00 |
| Noise | 183.90 | **149.60** | 248.90 | 180.95 | 842.55 | 173.50 | 150.50 | 155.65 | 842.55 | 842.55 |
| Confounder | 324.63 | 195.74 | 159.63 | 248.32 | 670.53 | 265.26 | 272.68 | 136.53 | **136.00** | 670.53 |
| Time-lag | 327.72 | 350.61 | 449.33 | 261.44 | 793.67 | 244.56 | 247.22 | **201.11** | 793.67 | 793.67 |
| Standardize | 228.32 | 201.79 | 349.63 | 243.84 | 840.26 | 244.42 | 310.63 | **159.84** | 840.26 | 840.26 |
| | | | | | **Climate** | | | | | |
| MAOOAM | 80.00 | 130.00 | 130.00 | 94.00 | **31.00** | 108.00 | 130.00 | 130.00 | 126.00 | 130.00 |
| ENSO | 529.36 | 530.27 | 453.00 | 589.36 | **337.09** | 666.27 | 608.73 | 665.36 | 666.27 | 666.27 |

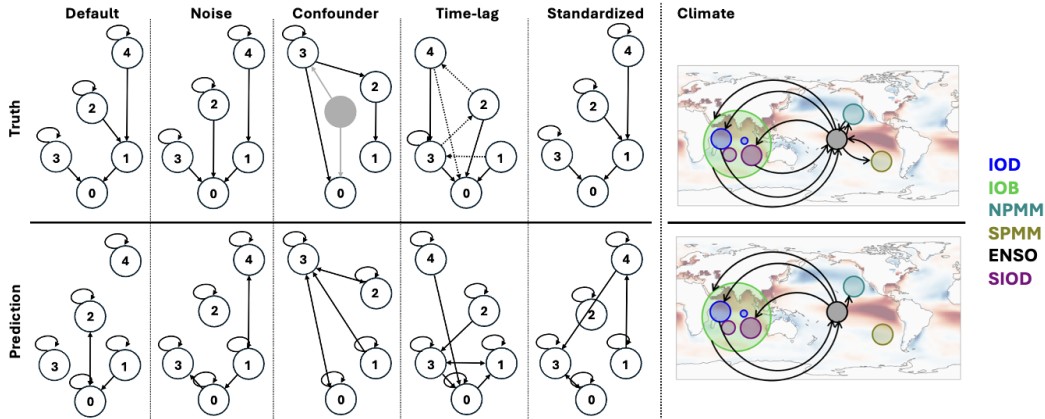

Figure 5: Example of baseline performance for coupled systems ($n = 5$) for causal challenges and ENSO in the decoupled Atlantic setting (climate). Inference is performed using the best performing algorithm. Grey nodes and edges represent unobserved confounder, and dashed lines denote time-lagged relationships.

as they generally only return absolute error counts independent from the edge density or graph size, thus making a comparison across graph hierarchies difficult.

# 6 Conclusion

We present **CausalDynamics**, an extensible data generation framework that we use to construct the largest benchmark dataset with over 14000 preprocessed graphs of increasing complexity. We structure our benchmark as a tiered system ranging from simple three-dimensional dynamical systems to pseudo-real physical systems. We provide a plug-and-play workflow to facilitate the development of novel causal discovery methods across various domains. Evaluating a set of state-of-the-art causal discovery algorithms on **CausalDynamics** shows that many advanced DL-based algorithms are outperformed by simpler methods, notably on high-dimensional datasets, highlighting a need for future method development. We believe our work provides the necessary foundation for the advancement of causal discovery algorithms that are applicable in high-dimensional, nonlinear and dynamical settings.

# Acknowledgment

We would like to thank Lars Lorch and Johannes Veith for discussions and their valuable insights. The authors acknowledge funding, computing, and storage resources from the NSF Science and Technology Center (STC) Learning the Earth with Artificial Intelligence and Physics (LEAP) (Award #2019625), and Department of Energy (DOE) Advanced Scientific Computing Research (ASCR) program (DE-SC0022255). JN also acknowledges funding from Columbia-Dream Sports AI Innovation Center. The project on which this report is based was funded by the Federal Ministry for Economic Affairs and Climate Action under the funding code 03EGTBW076. The responsibility for the content of this publication lies with the authors. Finally, the project was funded by the BW Pre-Seed program of the State of Baden-Württemberg, funded by the Ministry of Economics, Labour and Tourism and the L-Bank.

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

# CausalDynamics: A large-scale benchmark for structural discovery of dynamical causal models

## Supplementary Material

**Benjamin Herdeanu**[1,*], **Juan Nathaniel**[2,*], **Carla Roesch**[2,*],
**Jatan Buch**[2], **Gregor Ramien**[1], **Johannes Haux**[1], **Pierre Gentine**[2]

[1]kausable GmbH, [2]Columbia University

## Table of Contents

## A Getting started

We provide an overview, including code examples, on how to use the **CausalDynamics** Python package. The open-sourced code is available at `https://github.com/kausable/CausalDynamics/` and the latest documentation is published at `https://kausable.github.io/CausalDynamics/README.html`.

All results, code examples and descriptions rely on version `1.0.0` of the **CausalDynamics** Python package.

---

[*]Corresponding authors: benjamin@kausable.ai, jn2808@columbia.edu, cmr2293@columbia.edu

## A.1 Installation

The easiest way to install the Python package is via PyPi, see `https://pypi.org/project/causaldynamics/`, which currently requires `Python=3.10`.

```
$ pip install causaldynamics
```

It is also possible to install the package locally. Further installation instructions are available in the project repository at `https://kausable.github.io/CausalDynamics/README.html`.

## A.2 Download data

We provide a pre-generated dataset at `https://huggingface.co/datasets/kausable/CausalDynamics` that can be directly downloaded using the following commands:

```
$ wget https://huggingface.co/datasets/kausable/CausalDynamics/resolve
    /main/process_causaldynamics.py

$ python process_causaldynamics.py
```

The dataset was generated using the scripts published in the repository under: `https://github.com/kausable/CausalDynamics/tree/main/scripts`.

## A.3 Generate data

**Simple causal models**. Here, we showcase how to generate data for the simple complexity tier that consists of individual dynamical systems. As an example, we choose the Lorenz system and show how to (i) get the adjacency matrix of the system, (ii) solve the system for 1000 time steps resulting in time series trajectory data, and (iii) store the data.

```python
import xarray as xr
from causaldynamics.systems import get_adjacency_matrix_from_jac
from causaldynamics.systems import solve_system

# Define the system name, ...
system_name = "Lorenz"

# ... get the adjacency matrix of the system,...
A = get_adjacency_matrix_from_jac(sys_name)

# ... integrate the Lorenz ODE,...
data = solve_system(
    num_timesteps=1000,
    num_systems=1,
    system_name=system_name
)
data = xr.DataArray(data, dims=['time', 'node', 'dim'])

# ... and store the generated data
ds = create_dynsys_dataset(adjacency_matrix=A, time_series=data)
save_xr_dataset(ds, "out_path.nc)
```

We introduce function `solve_system`, a lightweight wrapper around the `dysts` package [60, 80], including ODE/SDE integration schemes that return the trajectories of the integrated system. The function `get_adjacency_matrix_from_jac` function then takes the Jacobian computed by `dysts` and extracts its adjacency matrix. For a visual walkthrough, see our explanatory notebook: `https://github.com/kausable/CausalDynamics/blob/main/notebooks/adj_mat_`

`from_dysts.ipynb`. Within our framework, a single dynamical system can correspond to a single root node, hence one of the dataset's dimension is referred to as node. Multiple systems can be simulated in parallel via the `num_systems` argument in `solve_system`.

Finally, we combine the generated time series (`data`) and its corresponding adjacency matrix $\mathcal{A}$ in a single dataset to save as a `NetCDF` file [100] using the `Xarray` package [101]. We followed the same procedure for the preprocessed benchmark data described in Appendix A.2.

More information and a detailed description of the simple causal model data generation can be found at `https://kausable.github.io/CausalDynamics/notebooks/simple_causal_models.html`.

**Coupled causal models.** We showcase example code of how to (i) create an SCM, (ii) simulate the system to generate the time series data, and (iii) plot the data. To create the SCM, we use the `create_scm` function which returns the corresponding adjacency matrix $\mathcal{A}$, the weights `W` and biases `b` of all MLPs and the `root_nodes` that act as temporal system drivers. We then simulate the system consisting of `num_nodes` nodes for `num_timesteps` driven by the dynamical systems `system_name` located on the root nodes. We provide a simple example for basic functionality and an advanced example that shows the modular plug-and-play feature configurability.

```python
# Simple example: Generate data of coupled causal models
from causaldynamics.creator import create_scm, simulate_system,
    create_plots

# Define system parameters
num_nodes = 2
node_dim = 3
num_timesteps = 1000
system_name='Lorenz'
confounders = False

# Create a coupled causal model,...
A, W, b, root_nodes, _ = create_scm(num_nodes, node_dim, confounders=
    confounders)

# ... simulate the system,...
data = simulate_system(A, W, b,
                       num_timesteps=num_timesteps,
                       num_nodes=num_nodes,
                       system_name=system_name)

# ... and plot the results.
create_plots(
            data,
            A,
            root_nodes=root_nodes,
            out_dir='.',
            show_plot=True,
            save_plot=False,
            create_animation=False,
        )
```

The `create_scm` and `simulate_system` function interface can be used to generate more complex graphs and causal challenges providing the option to introduce confounders, time-lag, noise, periodic functions as root nodes, and to standardize (i.e., correct for varsortability), which are visualized at `https://kausable.github.io/CausalDynamics/notebooks/causaldynamics.html`.

A more detailed introduction to coupled causal models with visualizations can also be found at `https://kausable.github.io/CausalDynamics/notebooks/coupled_causal_models.html`.

```python
# Advanced example: Generate more complex systems
from matplotlib import pyplot as plt
from causaldynamics.creator import create_scm, simulate_system
from causaldynamics.scm import create_scm_graph
from causaldynamics.plot import (
    plot_scm,
    plot_trajectories,
    plot_3d_trajectories
)
from causaldynamics.data_io import create_output_dataset,
    save_xr_dataset

# Define system parameters
num_nodes = 5
node_dim = 3
num_timesteps = 1000

confounders = False     # Set to True to add confounders
standardize = False     # Set to True to standardize the data
init_ratios = [1, 1, 1]# Set ratios of dynamical systems,
                        # periodic, and linear drivers at root nodes.
                        # Here: equal ratio.
system_name='random'    # Sample random dyn. sys. for root nodes
activations_names = ['identity', 'sin', 'sigmoid', 'tanh', 'relu']
                        # Activation names sampled uniformly per edge
noise = 0.5             # Set noise for the dynamical systems

time_lag = 10                       # Set time lag for time-lagged edges
time_lag_edge_probability = 0.1 # Set probability of time-lagged edges

# Create a coupled causal model,...
A, W, b, root_nodes, _ = create_scm(num_nodes,
                                    node_dim,
                                    confounders=confounders,
                                    time_lag=time_lag,
                                    time_lag_edge_probability=
                                        time_lag_edge_probability)

# ... simulate the system, ...
data = simulate_system(A, W, b,
                        num_timesteps=num_timesteps,
                        num_nodes=num_nodes,
                        system_name=system_name,
                        init_ratios=init_ratios,
                        time_lag=time_lag,
                        standardize=standardize,
                        activations_names=activations_names,
                        make_trajectory_kwargs={'noise': noise})

# ... visualize the results,...
plot_scm(G=create_scm_graph(A), root_nodes=root_nodes)
plot_3d_trajectories(data, root_nodes)
plot_trajectories(data, root_nodes=root_nodes, sharey=False)
plt.show()

# ... and save the data.
dataset = create_output_dataset(
        adjacency_matrix=A,
        weights=W,
        biases=b,
        time_lag=time_lag,
        time_series=data,
        root_nodes=root_nodes,
        verbose=False,
)
save_xr_dataset(dataset, "out_path.nc")
```

### A.4 Plotting functions

Plotting functions have been implemented to visualize the generated graph structures and system trajectories given the adjacency matrix $\mathcal{A}$, the time series `data` and optionally the `root_nodes` (see Figure 6).

```
# Plotting functions
from causaldynamics.scm import create_scm_graph
from causaldynamics.plot import (plot_scm, plot_trajectories,
    plot_3d_trajectories, animate_3d_trajectories)

# Plot SCM, root nodes are plotted in grey, dashed edges are time-
    lagged, dash-dotted edges as both, regular and time-lagged
plot_scm(G=create_scm_graph(A), root_nodes=root_nodes)

# Plot time-series at nodes, root nodes are plotted in grey
plot_trajectories(data, root_nodes=root_nodes, sharey=False)

# 3D plot of the systems at each node
plot_3d_trajectories(data, root_nodes, line_alpha=1.)

# 3D animation of the time-dependent trajectory buildup with rotating
    axes.
anim = animate_3d_trajectories(data,
                               root_nodes=root_nodes,
                               plot_type="subplots")
display(anim)
```

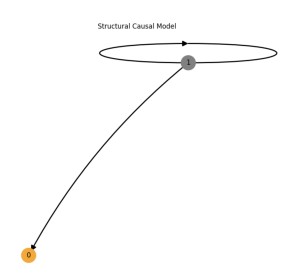

(a) Plot SCM for a graph consisting of 2 nodes (autocorrelated root node in grey).

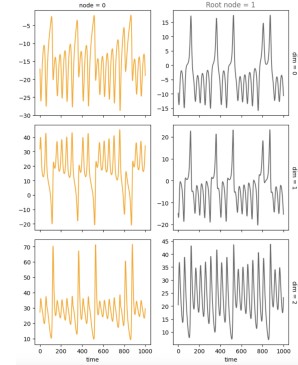

(b) Plotted trajectories for the SCM from (a).

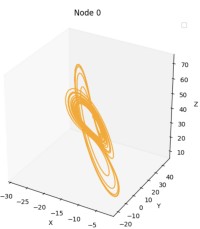 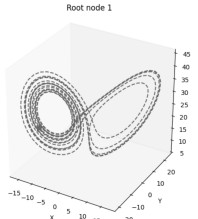

(c) 3D trajectories for the SCM from (a) and the time series from (b). These trajectories can also be animated to visualize their temporal development with rotating axes to intuitively grasp the 3D structure. The last frame of this animation is the visualized figure.

Figure 6: Plotting function in CausalDynamics at the example of a SCM consisting of 2 nodes, with the root node (grey) driven by a 3-dimensional dynamical system.

## A.5 Baseline evaluation

The following example demonstrates how to evaluate a single dynamical system. First, we load the dataset along with its generating causal graph (ground truth), represented as adjacency matrix.

```python
import xarray as xr
import copy
import numpy as np
from tqdm import tqdm
from causaldynamics.baselines import CUTSPlus

# Load dataset
ds = xr.open_dataset(DATA_DIR / "<SYSTEM_NAME>.nc")

# Extract timeseries and adjacency matrix as target
timeseries = ds['time_series'].to_numpy().transpose(1, 0, 2)
adj_matrix = ds['adjacency_matrix'].to_numpy()
```

For coupled systems, slight changes to data loading are required since there are several new adjacency matrices, such as the `adjacency_matrix_time_edges` for the lagged connection.

```python
# Select the first dimension of each multidimensional node
timeseries = ds['time_series'].to_numpy()[..., 0].transpose(1, 0, 2)

# `adjacency_matrix_summary` is the true adjacency matrix here
adj_matrix = ds['adjacency_matrix_summary'].to_numpy()
```

Finally, using `CUTS+` as an example, we provide sample evaluation script and visualize the predicted SCM in Figure 7.

```python
# Define parameters
tau_max = 1
corr_thres = 0.8

# Estimate adjacency matrix
cuts_adj_matrix = []
for x in tqdm(timeseries):

    # Initialize model
    cuts_model = CUTSPlus(tau_max=tau_max, corr_thres=corr_thres)

    # Fit the model
    cuts_model.run(X=x)

    # Extract inferred matrix
    cuts_adj_matrix.append(
        copy.deepcopy(cuts_model.adj_matrix)
    )

# Compute scores
score(preds= cuts_adj_matrix, labs= adj_matrix, name='CUTS+')

# Plot recovered SCM
G = create_scm_graph(cuts_model.adj_matrix)
plot_scm(G);
```

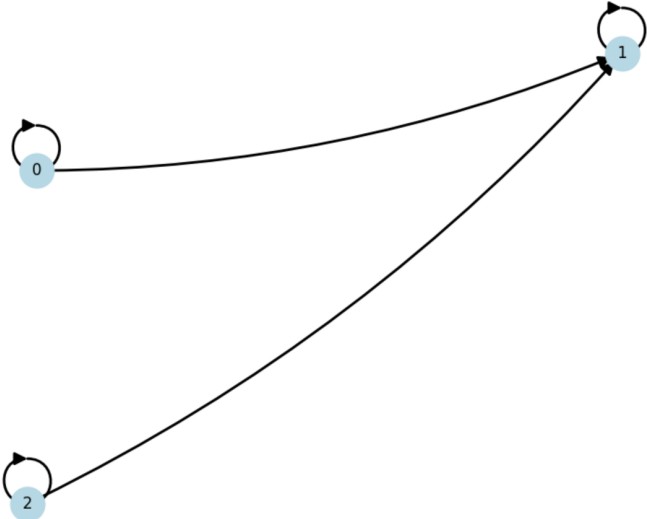

Figure 7: Predicted graph for a single dynamical system using CUTS+.

The complete baseline pipeline is available at `https://kausable.github.io/CausalDynamics/notebooks/eval_pipeline.html`.

We also provide the complete metrics for each graph in our HuggingFace data repository `https://huggingface.co/datasets/kausable/CausalDynamics`, but analyzing each of the 14k+ graphs can be too granular. As a result, we have added a built-in, easy-to-use diagnosis script that allows users to quickly analyze results on the experiment-level. For instance, if users are interested in understanding the effects of unobserved confounders (confounders=True) on the performance of different causal discovery algorithms, especially in light of high internal noise/stochasticity (noise=2.0), they can run the following:

```
# Diagnostic tool to evaluate each experiment
python diagnose.py --exp_dir data/simple/noise=2.00_confounder=True
```

# B Generate coupled causal models

Here, we elaborate on the propagation of information via MLPs in the DAG and provide the key algorithms used to generate coupled causal models.

## B.1 System initialization

To generate data for coupled causal models we need to initialize the number of time steps, the number of nodes (e.g., 5), the ratio of dynamical systems to periodic functions as root node drivers, the system name of dynamical system drivers, the dimensionality of the nodes (e.g., 3), and optional time-lags, as outlined in Algorithm 2 and sketched out in Figure 8.

If the dynamical system drivers are selected at random, the algorithm chooses randomly from the set of available 3D systems (see Algorithm 3). Users can also choose whether to include periodic or linear functions as drivers by initializing r appropriately. In this case, sinusoidal or linear trajectories are generated as outlined in Algorithm 4 and Algorithm 5. The generated data of the system's drivers is randomly assigned to the respective root nodes, and all non-root nodes initial states are set to zero (see Algorithm 6).

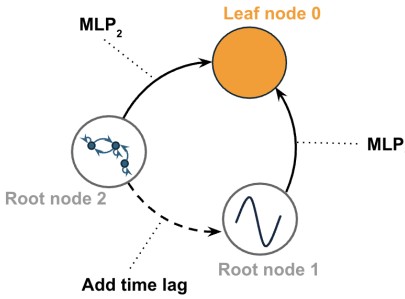

Figure 8: Initialize graph.

Finally, the initial time series data can be standardized in order to prevent varsortability as illustrated in Appendix B.3.

---

**Algorithm 2** Initialize Coupled Causal Models

---

**Require:** $T$ (num_timesteps), $N$ (num_nodes), $r$ (init_ratios), name (system_name), $D$ (node_dim), $l$ (time_lag)
**Ensure:** init $\in \mathbb{R}^{T \times N \times D}$ (initial values tensor)
 1: **if** $l > 0$ **then**
 2:     $T_{ext} \leftarrow T + l$                              ▷ Extended time for lag calculation
 3: **else**
 4:     $T_{ext} \leftarrow T$
 5: **end if**
 6: $n_{sys}, n_{sin}, n_{lin} \leftarrow$ allocate_elements_based_on_ratios$(N, r)$      ▷ Divide nodes by type
 7: **if** system_name = "random" **then**
 8:     $d_{sys} \leftarrow$ solve_random_systems$(T_{ext}, n_{sys})$     ▷ **Generate random system trajectories**
 9: **else**
10:     $d_{sys} \leftarrow$ solve_system$(T_{ext}, n_{sys},$ system_name$)$     ▷ Generate named system trajectories
11: **end if**
12: $d_{lin} \leftarrow$ drive_lin$(T_{ext}, n_{lin}, D)$                   ▷ **Generate linear trajectories**
13: $d_{sin} \leftarrow$ drive_sin$(T_{ext}, n_{sin}, D)$                 ▷ **Generate sinusoidal trajectories**
14: init $\leftarrow$ concat$(d_{sys}, d_{sin}, d_{lin}, \dim = 1)$              ▷ Combine trajectories
15: init $\leftarrow$ init$[:,$ randperm$(N), :]$                      ▷ Randomly permute nodes
16: **if** $l > 0$ **then**
17:     init_now $\leftarrow$ init$[: T, :, :]$                       ▷ Current time steps
18:     init_future $\leftarrow$ init$[l : T_{ext}, :, :]$        ▷ Future time steps for lagged edges
19:     init $\leftarrow$ concat(init_now, init_future, $\dim = 1$)       ▷ Combine current and future
20: **end if**
21: **return** init

---

**Algorithm 3** Generate random system trajectories

**Require:** $T$ (num_timesteps), $N$ (num_nodes)
**Ensure:** data $\in \mathbb{R}^{T \times N \times D}$ (trajectory tensor)
 1: names $\leftarrow$ get_3d_chaotic_system_names()                          ▷ Get list of available 3D systems
 2: s $\leftarrow$ random_sample(names, $\min(N, |\text{names}|)$)                          ▷ Select systems
 3: data $\leftarrow$ []                          ▷ Initialize empty trajectory list
 4: **for** $i \leftarrow 0$ to $N - 1$ **do**
 5:     sol $\leftarrow$ None
 6:     counter $\leftarrow 0$
 7:     **while** sol $=$ None **and** counter $<$ max_retry **do**                          ▷ Handle failing integration
 8:         system $\leftarrow$ random_choice(names $\setminus \{\text{s}[i \bmod |\text{s}|]\}$)
 9:         sol $\leftarrow$ solve_single_system(system, $T$)                          ▷ Solve system ODEs
10:         counter $\leftarrow$ counter $+ 1$
11:     **end while**
12:     **if** counter $\geq$ max_retry **then**
13:         **raise** Exception                          ▷ Failed to integrate system
14:     **end if**
15:     data.append(sol)                          ▷ Add trajectory to list
16: **end for**
17: data $\leftarrow$ convert_to_tensor(data)                          ▷ Convert to tensor
18: data $\leftarrow$ data.permute$(1, 0, 2)$                          ▷ Reshape to [T, N, D]
19: **return** data

---

**Algorithm 4** Generate Sinusoidal Driver Trajectories

**Require:** $T$ (num_timesteps), $N$ (num_nodes), $D$ (node_dim), $P_{max}$ (max_num_periods)
**Ensure:** data $\in \mathbb{R}^{T \times N \times D}$ (sinusoidal trajectory tensor)
 1: amplitude $\leftarrow 2 \times \text{rand}(N, D) - 1$                          ▷ Random amplitudes in $[-1, 1]$
 2: phase_shift $\leftarrow 2\pi \times \text{rand}(N, D)$                          ▷ Random phase shifts in $[0, 2\pi]$
 3: data $\leftarrow$ zeros$(T, N, D)$                          ▷ Initialize output tensor
 4: **if** $N > 0$ **then**
 5:     max_time $\leftarrow P_{max} \times 2\pi \times \text{rand}(N, D)$                          ▷ Random max times
 6:     time $\leftarrow$ linspace$(0, \text{max\_time}, T)$                          ▷ Generate time points
 7:     **for** $i \leftarrow 0$ to $T - 1$ **do**
 8:         data$[i, :, :] \leftarrow$ amplitude $\times \sin(\text{time}[i] + \text{phase\_shift})$                          ▷ Compute sine values
 9:     **end for**
10: **end if**
11: **return** data

---

**Algorithm 5** Generate Linear Driver Trajectories

**Require:** $T$ (num_timesteps), $N$ (num_nodes), $D$ (node_dim), $m_{\min}, m_{\max}$ (slope range), $b_{\min}, b_{\max}$ (intercept range)
**Ensure:** data $\in \mathbb{R}^{T \times N \times D}$ (linear trajectory tensor)
 1: $m \leftarrow (m_{\max} - m_{\min}) \times \text{rand}(N, D) + m_{\min}$                          ▷ Random slopes in $[m_{\min}, m_{\max}]$
 2: $b \leftarrow (b_{\max} - b_{\min}) \times \text{rand}(N, D) + b_{\min}$                          ▷ Random intercepts in $[b_{\min}, b_{\max}]$
 3: data $\leftarrow$ zeros$(T, N, D)$                          ▷ Initialize output tensor
 4: **if** $N > 0$ **then**
 5:     max_time $\leftarrow$ rand$(N, D)$                          ▷ Random max times in $[0, 1]$
 6:     time $\leftarrow$ linspace$(0, \text{max\_time}, T)$                          ▷ Generate time points
 7:     **for** $i \leftarrow 0$ to $T - 1$ **do**
 8:         data$[i, :, :] \leftarrow m \times \text{time}[i] + b$                          ▷ Compute linear values
 9:     **end for**
10: **end if**
11: **return** data

**Algorithm 6** Initialize state x for Coupled Causal Models

---

**Require:** init $\in \mathbb{R}^{T \times N \times D}$ (initial values tensor), $A \in \mathbb{R}^{N \times N}$ or $\mathbb{R}^{2N \times N}$ (adjacency matrix), standardize (boolean)
**Ensure:** $x \in \mathbb{R}^{T \times N \times D}$ (initialized tensor with values for root nodes and zero else)
1: $x \leftarrow$ zeros_like(init)                                    ▷ Initialize output tensor with zeros
2: **if** $A$.shape[0] $= A$.shape[1] **then**                        ▷ No time lag edges
3:    root_nodes $\leftarrow$ get_root_nodes_mask($A$)                 ▷ Identify nodes with no incoming edges
4:    $x[:, \text{root\_nodes}, :] \leftarrow \text{init}[:, \text{root\_nodes}, :]$    ▷ Set values only for root nodes
5: **else**                                                           ▷ With time lag edges
6:    $A_{now} \leftarrow A[: A.\text{shape}[0]/2]$                    ▷ Extract current time step connections
7:    $A_{past} \leftarrow A[A.\text{shape}[0]/2 :]$                   ▷ Extract past time step connections
8:    root_now $\leftarrow$ get_root_nodes_mask($A_{now}$)            ▷ Current time root nodes
9:    root_past $\leftarrow$ get_root_nodes_mask($A_{past}$)          ▷ Past time root nodes
10:   root_nodes $\leftarrow$ concat(root_now, root_past, dim $= 0$)  ▷ Combine masks
11:   $x[:, \text{root\_nodes}, :] \leftarrow \text{init}[:, \text{root\_nodes}, :]$    ▷ Set values only for root nodes
12: **end if**
13: **if** standardize **then**
14:   $\mu \leftarrow \text{mean}(x, \text{dim} = 0)$                  ▷ Compute mean over time
15:   $\sigma^2 \leftarrow \text{var}(x, \text{dim} = 0)$              ▷ Compute variance over time
16:   $x \leftarrow \frac{x - \mu}{\sqrt{\sigma^2}}$                   ▷ Standardize results
17: **end if**
18: **return** $x$

---

## B.2 MLP propagation

In the following, we outline how the information is passed through the DAG starting from the root nodes (see Figure 9). A detailed description of the MLP implementation to forward information through the DAG is giving in Algorithm 7.

Note, that this process follows the reverse order, i.e., from the highest node number to zero. This ensures that information is processed in the right order through the network because by construction, a node cannot have incoming edges from a node with higher degree (see Algorithm 1).

To correct for varsortability, we include the option to standardize the propagated time-series to maintain the variance of the original drivers along the causal structure (see Appendix B.3).

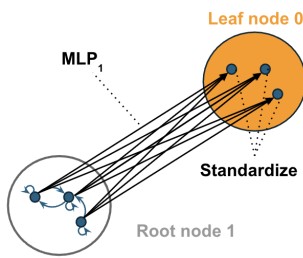

Figure 9: MLP propagation.

---

**Algorithm 7** MLP Propagation through the DAG

---

**Require:** $A \in \{0,1\}^{n \times n}$ (adjacency matrix), $W \in \mathbb{R}^{n \times d \times d}$ (weight tensor), $b \in \mathbb{R}^{n \times d}$ (bias tensor), init $\in \mathbb{R}^{T \times n \times d}$ (initial values), $\phi \in \mathcal{A}^n$, standardize $\in \{\text{True}, \text{False}\}$
**Ensure:** $x \in \mathbb{R}^{T \times n \times d}$ (propagated values)

1: $T, n, d \leftarrow \text{shape(init)}$        ▷ Get dimensions
2: $x \leftarrow \text{initialize\_x(init}, A)$        ▷ Initialize state tensor, see Algorithm 6
3: $A \leftarrow A^T$        ▷ Transpose adjacency matrix for easier indexing
4: **for** $i \leftarrow n-1$ to $0$ **do**        ▷ Process nodes in reverse topological order
5:      **if** $A.\text{shape}[0] \neq A.\text{shape}[1]$ **then**        ▷ Handle time lag edges
6:          $i \leftarrow i \bmod A.\text{shape}[0]$
7:      **end if**
8:      $m_i \leftarrow A[i].\text{bool}()$        ▷ Get incoming edge mask
9:      **if** $m_i.\text{any}()$ **then**        ▷ If node has any incoming edges
10:          $W_{sel} \leftarrow W[m_i]$        ▷ Select weights for incoming edges
11:          $b_{sel} \leftarrow b[m_i]$        ▷ Select biases for incoming edges
12:          $x_{sel} \leftarrow x[:, m_i]$        ▷ Select input values
13:          $y \leftarrow \text{matmul}(W_{sel}, x_{sel}) + b_{sel}$        ▷ MLP transformation
14:          $y \leftarrow \phi[i](y)$        ▷ Apply activation (optional)
15:          $y \leftarrow \text{sum}(y, \dim = 1)$        ▷ Aggregate incoming signals
16:          **if** standardize **then**
17:             $\mu \leftarrow \text{mean}(y, \dim = 0)$        ▷ Compute mean over time
18:             $\sigma^2 \leftarrow \text{var}(y, \dim = 0)$        ▷ Compute variance over time
19:             $y \leftarrow \frac{y - \mu}{\sqrt{\sigma^2}}$        ▷ Standardize results
20:          **end if**
21:          $x[:, i] \leftarrow x[:, i] + y$        ▷ Update node values
22:      **end if**
23: **end for**
24: **return** $x$

---

### B.3 Standardization

In this section we present our approach to standardize the generated data by estimating the mean and standard deviation over time in order to account for varsortability, following the idea of [24, 23]. Our approach is similar to standardizing data in for causal discovery on time series to account for confounders such as seasonality (e.g., [21, 9]). More details and examples can be found at `https://github.com/kausable/CausalDynamics/blob/main/notebooks/standardization.ipynb`.

As shown at the example of the SCM in Figure 10a, the variance increases with each node $v_k$ for $k = [0, 1, 2, 3, 4]$ (compare orange line in (b), and the corresponding time series in (c)). By standardizing the data following:

$$\tilde{x}_{v_k}(t) = \frac{x_{v_k}(t) - \mu_{v_k}}{\sigma_{v_k}}, \tag{8}$$

where $\mu_{v_k}$ and $\sigma_{v_k}$ are the mean and standard deviation over time, we can control for the variance increase (see blue line in (b), and the corresponding time series in (d)).

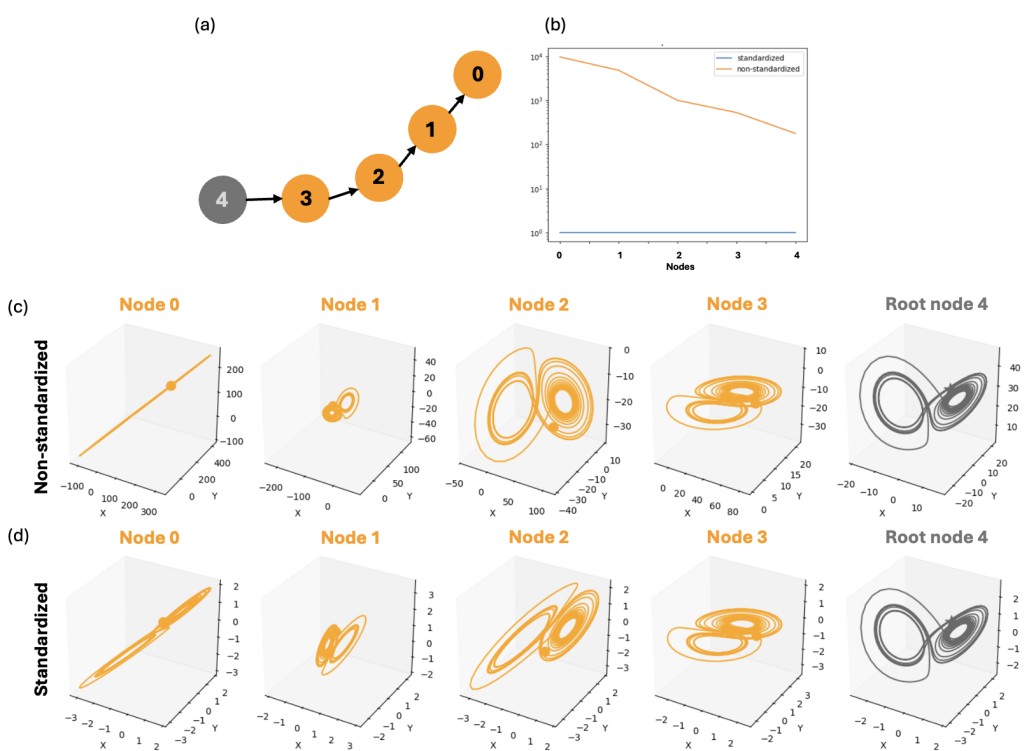

Figure 10: In (a) structural causal model consisting of 5 nodes. In (b) variance comparison for the non-standardized and standardized causal model along decreasing order of nodes (i.e., causal direction). In (c) trajectory of the non-standardized nodes and in (d) for the standardized nodes.

## C First-order climate models

In the following, we outline two pseudo-realistic examples of real world dynamical physical systems. For more details, please refer to the documentation at `https://kausable.github.io/CausalDynamics/notebooks/climate_causal_models.html`.

### C.1 XRO: eXtended nonlinear Recharge Oscillator model

The `XRO` model is a Python implementation [89] of a recharge oscillator (RO) model for El Niño-Southern Oscillation (ENSO) [91] coupled to stochastic-deterministic models for other climate modes (M) (as shown in Figure 11) which allows for a two-way interaction. A detailed description of the model is given by [92]. In the following we will provide an overview.

The system can be described through the following set of equations:

$$\frac{d}{dt}\begin{pmatrix} X_{\text{ENSO}} \\ X_{\text{M}} \end{pmatrix} = L \cdot \begin{pmatrix} X_{\text{ENSO}} \\ X_{\text{M}} \end{pmatrix} + \begin{pmatrix} N_{\text{ENSO}} \\ N_{\text{M}} \end{pmatrix} + \sigma_\xi \xi \tag{9}$$

with

$$\frac{d}{dt}\xi = -r_\xi \xi + w(t) \tag{10}$$

where $X_{\text{ENSO}} = [T_{\text{ENSO}}, h]$ and $X_{\text{M}} = [T_{\text{NPMM}}, T_{\text{SPMM}}, T_{\text{IOB}}, T_{\text{IOD}}, T_{\text{SIOD}}, T_{\text{TNA}}, T_{\text{ATL3}}, T_{\text{SASD}}]$ are the state vectors of averaged sea surface temperatures ($T$) over the respective regions (see Table 4). To describe the oscillatory behavior of ENSO the thermocline depth ($h$) averaged over the ENSO region is additionally included in $X_{\text{ENSO}}$.

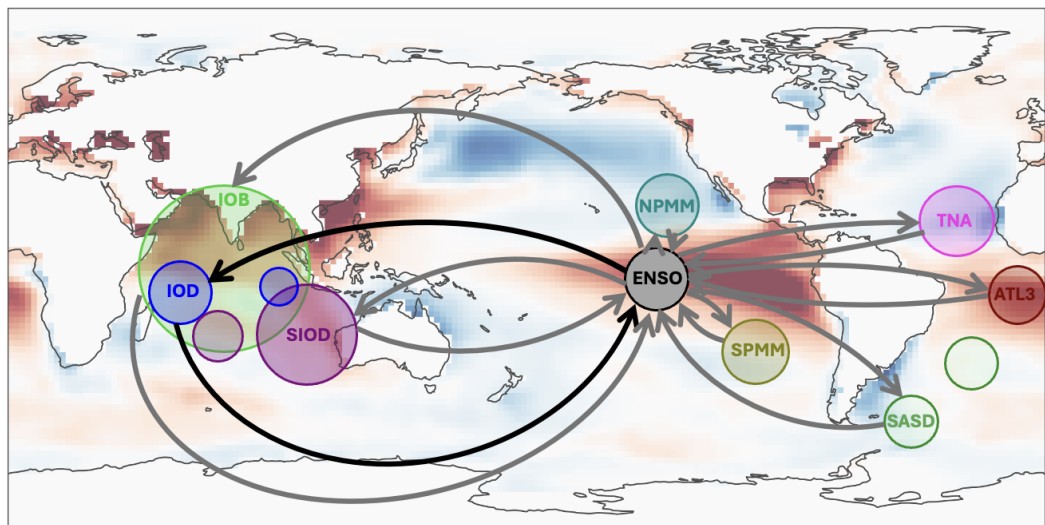

Figure 11: Anomalies of observed sea surface temperatures from the detrended ORAS5 reanalysis [102] for the period 1979-2019. Colored circles represent index regions for ENSO and other modes. Arrows visualize the associated causal graph with constantly coupled modes shown in black and potentially decoupled modes in grey. Figure inspired by [92].

Governing dynamics of $X = [X_{\text{ENSO}}, X_{\text{M}}]$ can be decomposed into linear ($L$), nonlinear ($N$) and stochastic ($\xi$) terms. Linear dynamics contain four submatrices:

$$L = \begin{pmatrix} L_{\text{ENSO}} & C_1 \\ C_2 & L_{\text{M}} \end{pmatrix} \tag{11}$$

where $L_{\text{ENSO}}$ describes ENSO internal dynamics, $L_{\text{M}}$ internal processes and interactions of other climate modes, and $C_1$ and $C_2$ represent the coupling matrices summarizing the feedback of ENSO on other modes and reversed, respectively. To conduct decoupling experiments we therefore intervene on $C_1$ and $C_2$. Note, that IOD cannot be decoupled as it is essential for the prediction of ENSO [103] (black arrows in Figure 11) and its asymmetry is therefore included in the nonlinear dynamics

$N$ along with quadratic terms describing ocean advection and sea surface temperature-wind stress feedbacks in the ENSO region [104, 105, 106, 107, 108]. Stochastic forcing $\xi$ is composed of weather and high-frequency noise for example the Madden-Julian Oscillation or westerly wind bursts. Due to a strong seasonality across climate modes, periodic parameters are added to the linear and nonlinear terms:

$$\boldsymbol{L} = \boldsymbol{L}_0 + \sum_{j=0}^{2} \left( \boldsymbol{L}_j^c \cos(j\omega t) + \boldsymbol{L}_j^s \sin(j\omega t) \right), \tag{12}$$

$$\boldsymbol{N} = \boldsymbol{N}_0 + \sum_{j=0}^{2} \left( \boldsymbol{N}_j^c \cos(j\omega t) + \boldsymbol{N}_j^s \sin(j\omega t) \right) \tag{13}$$

where $\omega = 2\pi/(12 \text{ months})$, and the subscripts $j = [0, 1, 2]$ refer to the mean, annual cycle and the semi-annual components, respectively.

Table 4: Definition of SST indices for climate modes used in the study.

| Climate Mode | Abbr. | Geographic region |
|---|---|---|
| El Niño–Southern Oscillation | ENSO | Niño3.4 region (170°–120°W, 5°S–5°N) |
| North Pacific Meridional Mode | NPMM | 160°–120°W, 10°–25°N |
| South Pacific Meridional Mode | SPMM | 110°–90°W, 25°–15°S |
| Indian Ocean Basin mode | IOB | 40°–100°E, 20°S–20°N |
| Indian Ocean Dipole mode | IOD | 50°–70°E, 10°S–10°N minus 90°–110°E, 10°S–0°N |
| Southern Indian Ocean Dipole mode | SIOD | 65°–85°E, 25°–10°S minus 90°–120°E, 30°–10°S |
| Tropical North Atlantic variability | TNA | 55°–15°W, 5°–25°N |
| Atlantic Niño | ATL3 | 20°W–0°E, 3°S–3°N |
| South Atlantic Subtropical Dipole | SASD | 60°–0°W, 45°–35°S minus 40°W–20°E, 30°–20°S |

## C.2 qgs: A flexible Python framework of reduced-order multiscale climate models

The qgs[†] library provides a Python implementation of a simplified, first-order coupled atmosphere–ocean model, based in part on the Modular Arbitrary-Order Ocean-Atmosphere Model (MAOOAM) [93]. It represents a two-layer quasi-geostrophic atmosphere ($\psi^1, \psi^3$), represented by barotropic $\psi_a$ and baroclinic $\theta_a$ streamfunctions, mechanically and thermally coupled to a shallow-water ocean component ($\psi_o$), with interactions driven by wind forcing alongside radiative and heat exchanges (see Figure 12a).

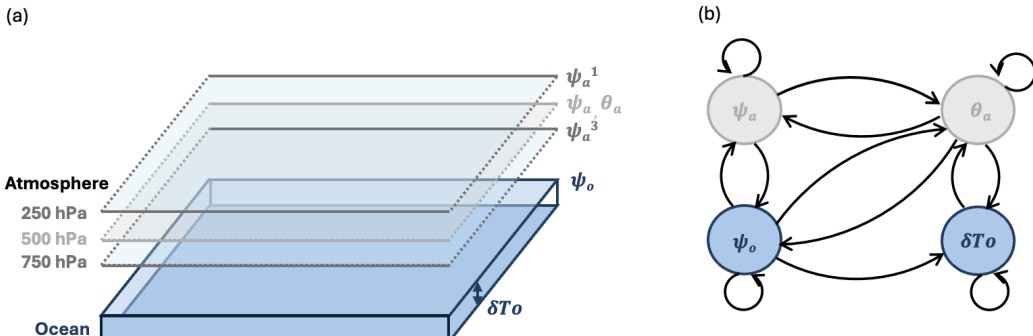

Figure 12: In (a) schematic of the atmosphere (grey) and ocean (blue) components of the simplified model, where dotted lines represent periodic and solid lines closed boundary conditions; figure inspired by `https://qgs.readthedocs.io/en/latest/files/model/maooam_model.html`. In (b) associated causal graph showing the coupling of the individual model components represented by the set of ODEs in Equations 14-17.

The core PDEs are the barotropic and baroclinic streamfunctions for the atmospheric layer and the ocean, plus anomalies of ocean $\delta T_o$ and atmospheric temperature $\delta T_a$. Each field is defined by a finite basis with a zonally periodic channel with no-flux boundary conditions in the meridional direction for the atmosphere and a closed basin with no-flux boundary conditions for the ocean. The fields are then projected on Fourier modes respecting these boundary conditions and the truncated model can be summarized by the following system of ODEs (see Figure 12b):

---

[†]`https://github.com/Climdyn/qgs/`

$$\dot{\psi}_{a,i} = -a_{i,i}^{-1} \sum_{j,m=1}^{n_a} b_{i,j,m}\left(\psi_{a,j}\psi_{a,m} + \theta_{a,j}\theta_{a,m}\right) - \beta\, a_{i,i}^{-1} \sum_{j=1}^{n_a} c_{i,j}\psi_{a,j}$$

$$- \frac{k_d}{2}\left(\psi_{a,i} - \theta_{a,i}\right) + \frac{k_d}{2}\, a_{i,i}^{-1} \sum_{j=1}^{n_o} d_{i,j}\psi_{o,j}, \tag{14}$$

$$\dot{\theta}_{a,i} = \frac{\frac{\sigma}{2}}{a_{i,i}\frac{\sigma}{2} - 1}\left\{ -\sum_{j,m=1}^{n_a} b_{i,j,m}\left(\psi_{a,j}\theta_{a,m} + \theta_{a,j}\psi_{a,m}\right) - \beta \sum_{j=1}^{n_a} c_{i,j}\theta_{a,j}\right.$$

$$\left. + \frac{k_d}{2}\, a_{i,i}(\psi_{a,i} - \theta_{a,i}) - \frac{k_d}{2}\sum_{j=1}^{n_o} d_{i,j}\psi_{o,j} - 2\,k_d'\, a_{i,i}\,\theta_{a,i}\right\}$$

$$+ \frac{1}{a_{i,i}\frac{\sigma}{2} - 1}\left\{ \sum_{j,m=1}^{n_a} g_{i,j,m}\,\psi_{a,j}\theta_{a,m} + (\lambda_a' + S_{B,a})\,\theta_{a,i}\right.$$

$$\left. - \left(\frac{\lambda_a'}{2} + S_{B,o}\right)\sum_{j=1}^{n_o} s_{i,j}\,\delta T_{o,j} - C_{a,i}'\right\}. \tag{15}$$

$$\dot{\psi}_{o,i} = \frac{1}{M_{i,i} + G}\left\{ -\sum_{j,m=1}^{n_o} C_{i,j,k}\,\psi_{o,j}\psi_{o,k} - \beta \sum_{j=1}^{n_o} N_{i,j}\psi_{o,j} - (d+r)\sum_{j=1}^{n_o} M_{i,j}\psi_{o,j}\right.$$

$$\left. + d \sum_{j=1}^{n_a} K_{i,j}\left(\psi_{a,j} - \theta_{a,j}\right)\right\}, \tag{16}$$

$$\delta\dot{T}_{o,i} = -\sum_{j,m=1}^{n_o} O_{i,j,m}\,\psi_{o,j}\,\delta T_{o,m} - (\lambda_o' + s_{B,o})\,\delta T_{o,i}$$

$$+ (2\lambda_o' + s_{B,a})\sum_{j=1}^{n_a} W_{i,j}\theta_{a,j} + \sum_{j=1}^{n_a} W_{i,j}C_{o,j}'. \tag{17}$$

Here, $W, K, d$ and $s$ denote the coupling coefficients governing ocean–atmosphere interactions; $a, g, b$ and $c$ are the inner-product coefficients for the atmospheric Fourier modes and $M, O, C, N$ those for the ocean. A detailed model description is given in [93] and `https://qgs.readthedocs.io/en/latest/files/model/maooam_model.html`.

# D  Benchmark details

## D.1  Summary

The final distribution of our preprocessed dataset is illustrated in Figure 13. For the SDE cases, we combine all dynamics with noise level $\delta > 0$. The nonlinear and periodic attributes refer to graphs where at least one of the root nodes represents only a simple dynamics or also mixed with periodic, nonlinear forcing.

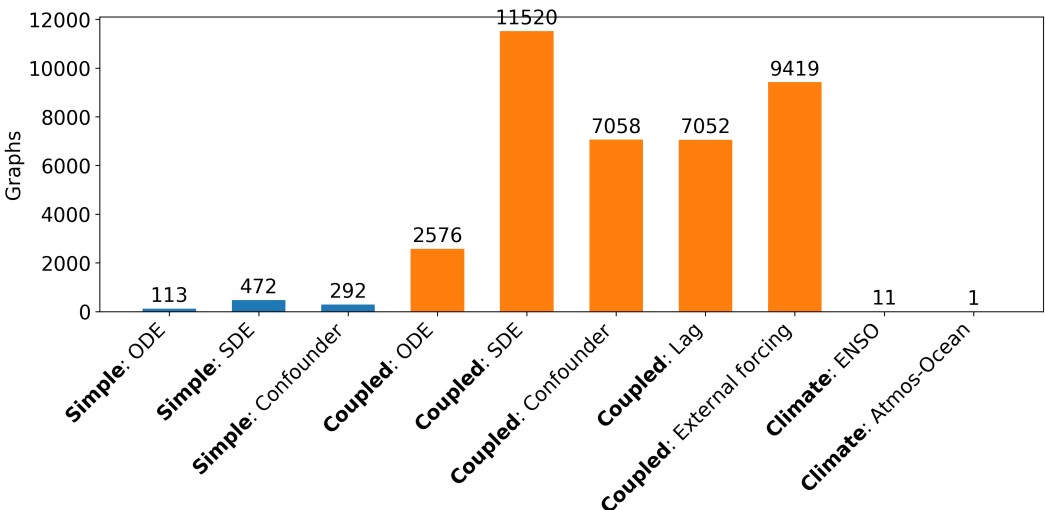

Figure 13: Distribution of graphs across hierarchy of dynamical systems, and their associated causal tasks meant to be solved.

## D.2  Hierarchy of Differential Equations

The data-generating code for the three tiers of dynamical system complexity described below can be found at `https://github.com/kausable/CausalDynamics/tree/main/scripts`.

**Simple**. From over 130 ordinary and stochastic models, we filter the `dysts` package[60] for three state variables and available Jacobian, which enables us to extract the adjacency matrix, yielding our 59 target systems. We generate different combination of graphs from these 59 uncoupled (simple) differential equations and perform the following combination of experiments: (i) varying the Langevin noise amplitude ($\delta \in \{0.0, 0.5, 1.0, 1.5, 2.0\}$, where $\delta > 0.0$ generates SDEs integrated with Euler-Maruyama integration scheme), and (ii) mimicking unobserved confounding scenarios. In total, 585 unique simple graphs are generated. Finally, the success rate for evaluating simple cases is $90\%$ (or 3612 out of 4095 possible evaluations). One of the most common failure reasons involves a singular value of the true adjacency matrix where score metrics such as AUROC fail.

**Coupled**. We then couple the simple systems and perform the following combination of experiments: (i) varying the Langevin noise amplitude ($\delta \in \{0.0, 0.5, 1.0, 1.5, 2.0\}$, (ii) mimicking unobserved confounding scenarios, (iii) changing the number of coupled nodes ($N_{node} \in \{3, 5, 10\}$), (iv) introducing periodic drivers (Coupled: Periodic) at equal ratio as dynamical system drivers or only having the latter, i.e., nonlinear chaotic drivers (Coupled: Nonlinear), (v) time-signal standardization, (vi) time-lags ($\tau \in \{0, 1\}$), (vii) nonlinear edge-level activations. Though 18000 coupled graphs are supposed to be generated, we keep 14096 of them, as the rest exhibit undesired outcomes, e.g., trajectories diverge, even after multiple retries.

**Climate.** We generated 11 graphs for the different coupling strategies in the coupled ENSO model, and 1 graph for the coupled atmosphere-ocean model. Specific hyparameters follow the original papers and discussed in Section C. All evaluation succeeded for the climate case.

# E   Experiment details

In this section, we describe additional experimental details, including the baseline models, hyperparameter choices, and evaluation metrics used. Unless otherwise stated, all experiments are conducted on 1xA100 NVIDIA GPU on a 100GB memory node with 12 CPU cores.

## E.1   Baselines

As part of our benchmark, we deploy 10 methods. We also summarize the final set of hyperparameters after a comprehensive search on a held-out dataset from the simple case, and attempt to set them to their default configurations. Nevertheless, we find that methods with little to no fine-tuning often outperform those that do so intensively. In all algorithms, we set the maximum lag $\tau_{\max} = 1$, unless otherwise specified. This choice is motivated by the fact that dynamical systems governed by differential equations typically evolve based on a single discretization step, which suffices to capture the system's causal structure in the resulting time series.

**PCMCI+** (Peter Clark Momentary Conditional Independence) [44] extends the PC-algorithm to multivariate time series by first preselecting candidate parents and then applying momentary conditional independence (MCI) tests across all lags to estimate contemporaneous and lagged edges under controlled false discovery. We set the maximum lagged time ($\tau_{max} = 1$) and the critical p-value ($\alpha_{crit} = 0.01$).

**F-PCMCI** (Filtered PCMCI) [73] extends PCMCI+ by incorporating transfer-entropy-based feature selection to prefilter candidate parents before momentary conditional independence testing, improving scalability and robustness in high-dimensional time series. We set the maximum lagged time ($\tau_{max} = 1$) and the critical p-value ($\alpha_{crit} = 0.01$).

**VARLiNGAM** [76] fits a vector autoregressive model (VAR) and then applies the LiNGAM (Linear Non-Gaussian Acyclic Model) [75] algorithm to the non-Gaussian residuals to recover a directed acyclic graph of contemporaneous effects alongside the estimated lagged coefficients. We set the maximum lagged time to $\tau_{max} = 1$.

**DYNOTEARS** [22] adopts a score-based structural VAR (SVAR) formulation to jointly estimate contemporaneous and time-lagged weight matrices by minimizing a penalized least-squares objective expended with a differentiable penalty that forbids any directed cycles in the instantaneous effects. We set the maximum lagged time to $\tau_{max} = 1$.

**TSCI** (Tangent Space Causal Inference) [79] builds on the idea that nonlinear dynamics locally resemble linear systems by estimating the data manifold's tangent spaces. TSCI first learns a continuous vector-field model for each variable's dynamics, e.g., via neural ODEs or Gaussian processes, then applies Convergent Cross Mapping (CCM) within each tangent to test for causal influence by assessing how well the state-space of one variable predicts another. This yields a single, interpretable causal graph for deterministic dynamical systems. We set the maximum lagged time ($\tau_{max} = 1$), the delay embedding dimension ($h_{embed} = 2$), and the correlation threshold ($\rho_{crit} = 0.8$).

**Neural GC** (Neural Granger Causality) [5] employs sparse-input multilayer perceptrons (cMLP) and long short-term memory (cLSTM) to model nonlinear autoregressive dependencies in multivariate time series, using input-weight regularization to identify directed Granger-causal links. In our evaluation, we focus solely on the cLSTM variant. We set the maximum lagged time ($\tau_{max} = 1$), number of hidden LSTM dimension ($h_{lstm} = 16$), with a learning rate of ($lr = 10^{-3}$) fitted over 100 epochs.

**CUTS+** [72] builds on the Granger-causality framework of CUTS [61] by combining a two-stage coarse-to-fine strategy with a message-passing graph neural network to discover causal structure in high-dimensional, irregularly-sampled time series. First, lightweight Granger-causality tests prefilter candidate parents for each variable, dramatically reducing the search space. Then, a GNN simultaneously imputes missing or unevenly spaced data and learns a sparse adjacency matrix via a penalized reconstruction loss that incorporates temporal encoding. This alternating imputation–learning loop makes CUTS+ both scalable and robust to missing data. We set the maximum lagged time ($\tau_{max} = 1$), the number of hidden MLP dimension ($h_{mlp} = 16$), the number of gated recurrent unit (GRU) ($n_{gru} = 1$), with a learning rate of $lr = 10^{-3}$ fitted over 10 epochs.

**RCD** (Repetitive Causal Discovery) [77] extends LiNGAM to settings with *latent confounders* under the linear, non-Gaussian, acyclic model (LiNGAM) assumption. Classic LiNGAM [75] identifies a unique DAG by exploiting non-Gaussianity and independence of error terms, but its basic form assumes causal sufficiency. RCD relaxes this by repeatedly partitioning variables into (approximate) causal orders and pruning edges using statistical tests, while allowing for latent common causes. Concretely, RCD alternates between: (i) *screening and ordering* steps that use correlation/independence diagnostics to propose parent sets under linear models with non-Gaussian residuals, and (ii) *refinement* steps that remove spurious links via independence checks on residuals (testing that, in the correct direction, residuals are independent of putative causes) and normality checks to ensure non-Gaussianity (a key LiNGAM identifiability condition). The repetitive loop iteratively updates ordering and adjacency until convergence, yielding a sparse adjacency estimate that is robust to latent confounding under the RCD model assumptions. We set the maximum lagged time ($\tau_{max} = 1$) and the critical p-value ($\alpha_{crit} = 0.01$).

**GRaSP** (Greedy Sparsest Permutations) [74] is a permutation-based causal discovery method that learns DAGs by searching over variable orderings. The method is grounded in the permutation-DAG correspondence: any variable ordering defines a unique minimal DAG consistent with the observed conditional independencies. Identifying the true DAG is then equivalent to finding the sparsest permutation, i.e., the one that yields the fewest edges. GRaSP employs a greedy local search strategy to approximate this combinatorial optimization. Starting from a random initial ordering, it iteratively applies adjacent swaps of variables. At each step, the algorithm updates the candidate DAG using conditional independence tests and accepts swaps that reduce the number of edges, gradually moving toward a sparser representation. This approach avoids the exhaustive search over all $n!$ permutations, making it scalable to moderately high-dimensional problems. By focusing on sparsity, GRaSP is particularly effective when the true underlying causal graph is relatively sparse and conditional independencies can be reliably tested. We set the maximum lagged time ($\tau_{max} = 1$).

**TCDF** (Temporal Causal Discovery Framework) [68] uses convolutional neural networks (CNN) with an attention mechanism to detect lagged causal relationships in multivariate time series, combining temporal filters and statistical pruning to handle nonlinear, high-dimensional, and noisy data. We set the maximum lagged time ($\tau_{max} = 1$), the CNN kernel size ($h_{cnn} = 4$), the dilation coefficient of 4 with a learning rate of $lr = 10^{-2}$ fitted over 100 epochs.

Note that most of these models, in the standard implementation, only infer a single and stationary causal graph. For this benchmark, we focus on summary graph estimation. Though some algorithms, such as PCMCI+, are able to infer a lagged adjacency matrix, this remains less common and will be the focus of the next version of the benchmark when the need arises and the choice of baseline algorithms capable of this proliferates.

## E.2 Metrics

We describe the metrics used in this work, including AUROC [94, 95], AUPRC [96], and SHD [97, 98].

Let $\mathcal{G} = (\mathcal{V}, \mathcal{E})$ be the true DAG on $n$ nodes $v_i \in \mathcal{V}$ with edge $(i, j) \in \mathcal{E}$ and let $\hat{s} : V \times V \to \mathbb{R}$ be a scoring function so that an edge $(i, j)$ is predicted whenever $\hat{s}(i, j) > \tau$. For each threshold $\tau$, we define a true-positive rate (TPR) and false-positive rate (FPR):

$$\text{TPR}(\tau) = \frac{\left|\{(i, j) \in \mathcal{E} : \hat{s}(i, j) > \tau\}\right|}{|\mathcal{E}|}, \quad \text{FPR}(\tau) = \frac{\left|\{(i, j) \notin \mathcal{E} : \hat{s}(i, j) > \tau\}\right|}{n(n-1) - |\mathcal{E}|}, \quad (18)$$

The Area Under the receiver operating characteristic curve (AUROC) is then:

$$\text{AUROC} = \int_0^1 \text{TPR}\big(\text{FPR}^{-1}(u)\big)\, du, \quad (19)$$

which defines the probability that a true edge ranks above a false one, i.e., 0.5 means random, $< 0.5$ means worse than random, 1.0 means a perfect prediction.

Similarly, we define precision and recall:

$$\text{Precision}(\tau) = \frac{\left|\{(i, j) \in \mathcal{E} : \hat{s}(i, j) > \tau\}\right|}{\left|\{(i, j) : \hat{s}(i, j) > \tau\}\right|}, \quad \text{Recall}(\tau) = \text{TPR}(\tau). \quad (20)$$

The area under the precision–recall curve (AUPRC) is:

$$\text{AUPRC} = \int_0^1 \text{Precision}\big(\text{Recall}^{-1}(r)\big)\, dr. \tag{21}$$

which unlike AUROC depends on the edge-density $\frac{\mathcal{E}}{n(n-1)}$ and thus more accurately reflects performance in the sparse-graph regime. For AUROC the optimal score is 1, i.e., perfect prediction.

Finally, SHD is defined as the minimum number of edge additions, deletions, or reversals required to transform $\hat{\mathcal{G}}$ into $\mathcal{G}$:

$$\text{SHD}(\hat{\mathcal{G}}, \mathcal{G}) = \big|\{(i,j) \in \mathcal{E} \setminus \hat{\mathcal{E}}\}\big| + \big|\{(i,j) \in \hat{\mathcal{E}} \setminus \mathcal{E}\}\big| + \#\{(i,j) : (i,j) \in \hat{\mathcal{E}}, (j,i) \in \mathcal{E}\}. \tag{22}$$

which a smaller SHD indicates a closer match to the ground truth, with 0 corresponding to exact recovery. Unlike AUROC and AUPRC, which are ranking-based metrics, SHD directly evaluates the correctness of the learned graph structure.

# F  Additional results

## F.1  Time discretisation ablation

We investigate the sensitivity of causal discovery algorithms to different sampling frequencies by sub-sampling every 1st, 5th, or 10th time step. As an illustration, we consider one of the most difficult experiments: coupling=nonlinear_noise=2.00_systems=10_confounder=True_standardize=True_timelag=1.
Results are shown in Table 5. Overall, we find that reducing sampling frequency tends to improve detection accuracy. This is potentially due to the reduction of data redundancy that can noise the algorithm.

Table 5: Ablating time discretisation: AUROC, AUPRC, and SHD scores at different sampling frequencies (every 1st, 5th, or 10th step).

| Model | AUROC (1/5/10) | AUPRC (1/5/10) | SHD (1/5/10) |
|---|---|---|---|
| PCMCI+ | 0.486/0.550/0.553 | 0.382/0.415/0.425 | 51.9/44.4/40.5 |
| F-PCMCI | 0.547/0.557/0.566 | 0.406/0.418/0.429 | 45.7/44.2/38.9 |
| VARLiNGAM | 0.519/0.518/0.548 | 0.389/0.389/0.407 | 51.9/51.3/47.2 |
| DYNOTEARS | 0.546/0.539/0.574 | 0.426/0.416/0.444 | 38.9/41.6/37.9 |
| NGC-LSTM | 0.497/0.500/0.500 | 0.377/0.378/0.378 | 56.0/34.0/34.0 |
| TSCI | 0.533/0.576/0.567 | 0.416/0.458/0.443 | 41.4/38.6/36.4 |
| CUTS+ | 0.495/0.500/0.500 | 0.376/0.378/0.378 | 39.0/34.0/34.0 |
| RCD | 0.500/0.501/0.505 | 0.378/0.382/0.385 | 34.0/34.1/34.1 |
| GRaSP | 0.498/0.484/0.490 | 0.385/0.379/0.384 | 35.8/36.2/36.2 |
| TCDF | 0.500/0.500/0.500 | 0.378/0.378/0.378 | 34.0/34.0/34.0 |

## F.2  Partial observability ablation

We also test algorithm performance under partial versus full observability, by sampling only one variable per multidimensional node versus averaging across all variables at each node. As an illustration, we consider one of the most difficult experiments: coupling=nonlinear_noise=2.00_systems=10_confounder=True_standardize=True_timelag=1. Results are shown in Table 6. We find that fully observing the system yields a modest improvement in detection performance. One possible explanation is that, even when some variables are hidden, their influence still propagates through the variables we do observe, so the loss of information is less severe than it might appear.

Table 6: Ablating partial observability: AUROC, AUPRC, and SHD scores for partially vs. fully observed systems.

| Model | AUROC (partial/full) | AUPRC (partial/full) | SHD (partial/full) |
|---|---|---|---|
| PCMCI+ | 0.486/0.500 | 0.382/0.386 | 51.9/50.2 |
| F-PCMCI | 0.545/0.546 | 0.405/0.406 | 45.8/47.4 |
| VARLiNGAM | 0.519/0.503 | 0.389/0.381 | 51.9/51.7 |
| DYNOTEARS | 0.546/0.559 | 0.426/0.426 | 38.9/37.2 |
| NGC-LSTM | 0.499/0.500 | 0.377/0.378 | 55.9/34.0 |
| TSCI | 0.533/0.552 | 0.416/0.421 | 41.4/38.5 |
| CUTS+ | 0.496/0.500 | 0.376/0.378 | 39.2/34.0 |
| RCD | 0.500/0.499 | 0.378/0.377 | 34.0/34.0 |
| GRaSP | 0.498/0.499 | 0.385/0.388 | 35.8/36.4 |
| TCDF | 0.500/0.500 | 0.378/0.378 | 34.0/34.0 |

## F.3  Edge-level activation ablation

We also perform additional experiment where the edge-level activation functions vary. For each identical `experiment_id`, we pass a new argument `activation=mixed`, which by default ran-

domly samples `activations_names=["identity", "sin", "sigmoid", "tanh", "relu"]` as the graph's edge-level activation. We compare results for the following experiments with coupling=nonlinear_noise=0.00_systems=3_confounder=False_standardize=False_timelag=0:

1. Linear activation (identity)
2. Mixed activation

Table 7: Ablating different (linear vs. mixed nonlinear) activations: AUROC, AUPRC, and SHD scores.

| Model | AUROC (linear / mixed) | AUPRC (linear / mixed) | SHD (linear / mixed) |
|---|---|---|---|
| CUTS+ | 0.500 / 0.478 | 0.378 / 0.417 | 37.200 / 45.667 |
| DYNOTEARS | 0.532 / 0.646 | 0.407 / 0.541 | 39.500 / 30.167 |
| F-PCMCI | 0.546 / 0.604 | 0.405 / 0.493 | 46.650 / 36.333 |
| GRaSP | 0.521 / 0.488 | 0.396 / 0.437 | 35.500 / 42.667 |
| NGC-LSTM | 0.498 / 0.503 | 0.377 / 0.428 | 55.900 / 51.278 |
| PCMCI+ | 0.526 / 0.560 | 0.398 / 0.462 | 47.900 / 41.167 |
| RCD | 0.500 / 0.495 | 0.378 / 0.424 | 34.000 / 38.333 |
| TCDF | 0.500 / 0.491 | 0.378 / 0.430 | 34.000 / 41.167 |
| TSCI | 0.537 / 0.689 | 0.414 / 0.601 | 39.850 / 24.889 |
| VARLiNGAM | 0.503 / 0.554 | 0.380 / 0.458 | 51.700 / 43.667 |

We find that mixed activations tend to improve causal discovery performance, potentially because directionality is more detectable in nonlinear cases, i.e., the signal is less invertible (see LiNGAM paper [75]). This effect is especially pronounced for deep learning–based methods such as TSCI, which shows an improvement of the SHD score by 40% (Table 7). This suggests that DL-based methods are better at capturing nonlinearities in the data.

