# OpenReview forum: "CausalDynamics: A large‐scale benchmark for structural discovery of dynamical causal models"
_NeurIPS.cc/2025/Datasets_and_Benchmarks_Track — NeurIPS 2025 Datasets and Benchmarks Track poster_

### Official Review · Reviewer_Vt55 · 2025-06-18

**Rating:** 5
**Confidence:** 3

**Summary:**

The paper introduces CausalDynamics, a benchmark for evaluating causal discovery algorithms on time series data. It consists of three types of problems with varying degrees of difficulty. The easiest ones describe three-dimensional chaotic systems. The second tier takes such systems as well as periodic functions and combines them into larger graphs. Lastly, the third tier consists of two real-world problems on climate data and atmosphere-ocean data. The paper evaluates several causal discovery algorithms on the benchmark.

**Additional Feedback:**

Questions

1. What is the reason for using the Growing Network with Redirection model?

2. Confounding in the coupled case is not entirely clear to me yet. Why does a 90 degree rotation result in confounded graphs? Also, should the right edge in Figure 3 (b) be black, considering the array to the left of it? I think I did not understand something correctly here.

3. How representative are chaotic systems for time series in real-world applications? Should causal discovery algorithms that perform well on this benchmark also perform well on most other real-world problems? In other words, is this benchmark supposed to evaluate causal discovery algorithms on time series in general, or only with respect to a specific type of problem (that is related to chaotic systems)?

4. Is there a connection between the third tier (real-world benchmarks) and the first two? How do they connect to the benchmark? And what is their contribution in this paper, compared to how they were published originally?

5. Can you please elaborate on Figure 4? What do the x-axis and y-axis show?

Minor points

1. Some figures in the appendix should be improved. Figure 10(b) has a very small legend and axis description. In Figure 13, one number intersects with the horizontal line at the top.

2. In line 132: "Further, even though datasets might contain thousands of samples [15], the underlying graphs are but a handful." What does this mean exactly? Does it mean that these datasets are large but only based on a small number of causal graphs?

3. The paper sometimes mentions this benchmark being "the largest benchmark". What exactly does the word "largest" refer to here? Largest with respect to what?

4. I think the comma in line 283 should be removed.

**Dataset Code Accessibility:**

Yes

**Dataset Code Comments:**

I loaded the files from Hugging Face and did not find any problems with them.

**Ethical Considerations:**

No, there are no or only very minor ethics concerns

**Final Justification:**

The authors expanded the benchmark in several ways, taking feedback from me and other reviewers into account. I believe that the resulting benchmark is a valuable contribution to the community and recommend acceptance.

**Limitations Weaknesses:**

1. The Simple and Coupled datasets are mostly created from chaotic systems. These datasets are valuable, however, the paper does not discuss the limitations with respect to this.
2. In Coupled, all edges are created by summing over different MLP outputs. This results in very specific functions, and is not necessarily representative of a lot of real-world relations.

**Strengths Contributions:**

1. The paper is well-written and clearly understandable.
2. It includes a large number of datasets for causal discovery in dynamical systems, with different levels of difficulty.
3. Multiple options and scenarios are included, such as confounding, lag, and standardization.
4. Two real-world datasets are included in the benchmark.
5. The experimental evaluation covers multiple causal discovery algorithms.

---

> ### Author Rebuttal · Authors · 2025-07-30
>
> We thank reviewer Vt55 for the valuable comments that contributed to improve this work. In the following we will provide targeted responses to each of the reviewer’s comments.
>
> > The Simple and Coupled datasets are mostly created from chaotic systems. These datasets are valuable, however, the paper does not discuss the limitations with respect to this.
>
> We have now added a clarification how this might impact the generalizability of algorithm performance to more stable or weakly nonlinear systems. We further added a clarification that more stable systems can be generated by simply using the periodic / newly implemented linear coupling functions or lower dimensional attractors dynamics can be generated by discarding initial time steps (e.g. the first 500) that are less dependent on the initial conditions. Finally, the structure of our framework is not just representative of the climate system but also applies to adjacent fields (e.g. [1]).
>
> > In Coupled, all edges are created by summing over different MLP outputs. This results in very specific functions, and is not necessarily representative of a lot of real-world relations.
>
> In order to further capture the complex real-world relationships, we have implemented several activation functions `("identity", "sin", "sigmoid", "tanh", "relu")` where users can mix-and-match randomly initialized edge-level activations. Given the hierarchical nature of our benchmark, stacking several nonlinearly-activated MLPs can, in principle, produce any complex interactions (i.e., neural network as a universal function approximator, more discussion below). Updated API is shown here and described in the updated documentation:
>
> ```
> da = simulate_system(
> 	…,
>         activations_names = ["identity", "sin", "sigmoid", "tanh", "relu"]
> )
> ```
>
> For instance, `sin` activation often relates to systems with inherent cyclicity (e.g., climate), `"sigmoid"/"tanh"` are often adopted in signal processing (e.g., digital signals with binary thresholding in remote sensing, medical equipment, sensors). And a mixture of them can be used at once to mimic extremely complex, highly nonlinear relationships.
>
> Now, for each identical `experiment_id`, we pass a new argument, `activation=mixed`, which by default specifies `activations_names=["identity", "sin", "sigmoid", "tanh", "relu"]`. A comparison of result with 20 randomly initialized graphs with the following `experiment_ids`:
> - No activation (linear): `coupling=nonlinear_noise=0.00_systems=3_confounder=False_standardize=False_timelag=0_activation=none`
> - Mixed activation (linear and non-linear): `coupling=nonlinear_noise=0.00_systems=3_confounder=False_standardize=False_timelag=0_activation=mixed`
>
> Note that we have **added 3 more baseline algorithms**, including GRaSP [2], RCD [3], and TCDF [4], as well as an **additional metric**, SHD [5].
>
> *Table 1: Ablating different (linear/non-linear) activations*
>
> |Model|AUROC (linear/mixed)|AUPRC (linear/mixed)|SHD (linear/mixed)|
> |--|--|--|--|
> |cutsplus|0.500 / 0.478|0.378 / 0.417| 37.200 / 45.667|
> |dynotears|0.532 / 0.646 |0.407 / 0.541|39.500 / 30.167|
> |fpcmci|0.546 / 0.604 |0.405 / 0.493|46.650 / 36.333|
> |grasp|0.521 / 0.488|0.396 / 0.437|35.500 / 42.667|
> |ngclstm|0.498 / 0.503|0.377 / 0.428|55.900 / 51.278|
> |pcmciplus|0.526 / 0.560|0.398 / 0.462|47.900 / 41.167|
> |rcd|0.500 / 0.495|0.378 / 0.424|34.000 / 38.333|
> |tcdf|0.500 / 0.491 | 0.378 / 0.430 | 34.000 / 41.167|
> |tsci|0.537 / 0.689|0.414 / 0.601|39.850 / 24.889|
> |varlingam|0.503 / 0.554|0.380 / 0.458|51.700 / 43.667|
>
> We find that mixed activations tend to improve causal discovery model performance, potentially because directionality tends to be more detectable in nonlinear cases, i,e., the signal is less invertible (see LiNGAM paper [6]). This is especially the case for deep learning-based (DL) methods such as TSCI which shows an improvement of the SHD score by 40% (see Table 1). This is potentially due to DL-based methods capturing the nonlinearities better.
>
> While a 3-layer neural network with nonlinear activations is theoretically a universal function approximator [7], **real-world systems** are often far more complex than the SCMs used in our benchmark. Nonetheless, these benchmarks already pose significant challenges to current causal discovery methods, despite the advantage of having known ground truth graphs by design. For objective evaluation, access to the true causal structure is essential. In real-world scenarios, this is rarely achievable due to confounding influences and the system's embedding with its environment. In summary, synthetic datasets remain crucial for benchmarking causal discovery algorithms, as they provide the only setting where performance can be rigorously and fairly assessed as the ground truth is known.
>
> **Additional Questions**
>
> > What is the reason for using the Growing Network with Redirection model?
>
> We chose the GNR architecture as it allows for the easy implementation of causal challenges, such as time lag, nonlinearity or hidden confounding. Further, scale-free graphs have shown to be representative of many real-world systems.
>
> > Confounding in the coupled case is not entirely clear to me yet. Why does a 90 degree rotation result in confounded graphs? Also, should the right edge in Figure 3 (b) be black, considering the array to the left of it? I think I did not understand something correctly here.
>
> Rotating the adjacency matrix by 90 degrees (excluding the diagonal), we can add an additional link (see green array in Fig. 3b) while guaranteeing to maintain the rest of the original graph structure. Yes, the right edge should be black. We have now updated Figure 3b.
>
> > How representative are chaotic systems for time series in real-world applications? Should causal discovery algorithms that perform well on this benchmark also perform well on most other real-world problems? In other words, is this benchmark supposed to evaluate causal discovery algorithms on time series in general, or only with respect to a specific type of problem (that is related to chaotic systems)?
>
> Many real-world systems are chaotic (e.g., climate systems), while others are highly nonlinear and may exhibit chaotic behavior (e.g., ecology, financial markets). More stable dynamics can be represented in the benchmark by discarding initial time steps (e.g., the first 500) allowing the trajectories to settle around fixed points, if any.
> Furthermore, CausalDynamics focuses on time series data. Our goal is to evaluate causal discovery algorithms under conditions that closely represent real-world complexity (e.g., chaotic, noisy, confounded). Strong performance on the benchmark is expected to correlate with real-world applicability since the design of the benchmark includes a diverse range of challenges, enabling a more robust assessment of algorithm performance across different systems.
>
> > Is there a connection between the third tier (real-world benchmarks) and the first two? How do they connect to the benchmark? And what is their contribution in this paper, compared to how they were published originally?
>
> The two models in Tier 3 were originally developed to study common climate phenomena, and their well-known reduced-order representations are the Lorenz systems (Tier 1 and 2). In future work, the framework can be extended to other domains where their simpler representations can be traced back to Tier 1 and 2 counterparts (e.g., Lorenz to represent climate systems and Rössler kinetics in Chemistry).
>
> > Can you please elaborate on Figure 4? What do the x-axis and y-axis show?
>
> X-axis values show the CDF, while the y-axis represents the respective AUROC and AUPRC scores. We realise that this is not clear from the figure and have elaborated this now in the figure caption.
>
> > Some figures in the appendix should be improved. Figure 10(b) has a very small legend and axis description. In Figure 13, one number intersects with the horizontal line at the top.
>
> > In line 132: "Further, even though datasets might contain thousands of samples [15], the underlying graphs are but a handful." What does this mean exactly? Does it mean that these datasets are large but only based on a small number of causal graphs?
>
> > The paper sometimes mentions this benchmark being "the largest benchmark". What exactly does the word "largest" refer to here? Largest with respect to what?
>
> > I think the comma in line 283 should be removed.
>
> Thank you for these points. We have corrected points 2 and 4 in the manuscript and further improved Figures 10 and 13. Regarding point 3, we refer to the number of causal graphs. In our case, this is > 15k graphs.
>
> References:
>
> [1] S. M. Smith, K. L. Miller, G. Salimi-Khorshidi, M. Webster, C. F. Beckmann, T. E. Nichols, J. D. Ramsey, and M. W. Woolrich. *Network modelling methods for FMRI*. Neuroimage, 54 (2):875–891, 2011.
>
> [2] Wai-Yin Lam, Bryan Andrews, and Joseph Ramsey. *Greedy relaxations of the sparsest permutation algorithm*. In Uncertainty in Artificial Intelligence, 1052–1062. PMLR, 2022.
>
> [3] Takashi Nicholas Maeda, Shohei Shimizu. *RCD: Repetitive causal discovery of linear non-Gaussian acyclic models with latent confounders*. In Proceedings of the Twenty Third International Conference on Artificial Intelligence and Statistics, PMLR 108:735-745, 2020.
>
> [4] Nauta M, Bucur D, Seifert C. *Causal Discovery with Attention-Based Convolutional Neural Networks*. Machine Learning and Knowledge Extraction. 2019; 1(1):312-340.
>
> [5] Ioannis Tsamardinos, Laura E Brown, and Constantin F Aliferis. *The max-min hill-climbing bayesian network structure learning algorithm*. Machine Learning, 65(1):31–78, 2006.
>
> [6] Shimizu, S., Hoyer, P. O., Hyvärinen, A., Kerminen, A., & Jordan, M. (2006). *A linear non-Gaussian acyclic model for causal discovery*. Journal of Machine Learning Research, 7(10).
>
> [7]  Hornik, K., Stinchcombe, Maxwell; White, Halbert (1989). *Multilayer feedforward networks are universal approximators*. Neural Networks. 2 (5): 359–366.

---

> > ### Comment · Reviewer_Vt55 · 2025-08-04
> >
> > I thank the authors for their detailed and extensive rebuttal. I like the changes they made to their benchmark, both in response to my and the other authors' reviews. I do not have any further questions and will increase my score accordingly.

---

### Official Review · Reviewer_Xssv · 2025-06-27

**Rating:** 5
**Confidence:** 4

**Summary:**

The paper introduces CausalDynamics, a new, large-scale benchmark and extensible data generation framework designed to advance causal discovery for complex dynamical systems. The authors argue that existing benchmarks are insufficient, as they are often tailored to deterministic, low-dimensional, or static systems, failing to capture the challenges inherent in real-world physical processes.

The framework generates data across three tiers of increasing complexity: simple, 3D chaotic dynamical systems; hierarchically coupled systems created via a novel graph generation algorithm, allowing for the controlled introduction of challenges like confounding, noise, and time-lags; and pseudo-real data from two idealized atmosphere-ocean climate models.

The authors adapt an existing algorithm (GNR) to couple various dynamical systems (e.g., chaotic ODEs, periodic functions) via MLPs, enabling the creation of complex, hierarchical causal structures with specified properties.

**Additional Feedback:**

This is an excellent paper and a valuable contribution to the community. My main suggestion is to consider a deeper diagnostic evaluation in future work. While the current paper successfully establishes the benchmark and provides a strong initial analysis, the real value will be unlocked by using it to understand the specific failure modes of existing algorithms.

Maybe just out of curious, does this paper: Benchmarking Structural Inference Methods for Interacting Dynamical Systems with Synthetic Data, (accepted by NeurIPS 2024 Dataset and Benchmark track) share similar research interest with this submission?

**Dataset Code Accessibility:**

Yes

**Dataset Code Comments:**

Yes, the submission is readily accessible, available in a final and usable format, and is exceptionally well-documented, making it highly reproducible.

**Ethical Comments:**

The paper includes a comprehensive NeurIPS checklist, and I concur with the authors' self-assessment that no significant ethical concerns remain.

**Ethical Considerations:**

No, there are no or only very minor ethics concerns

**Final Justification:**

Dear AC, I will keep my rating as 5: Accept. I do believe this submission would be a good addition to the research community.

**Limitations Weaknesses:**

1. The evaluation of SOTA methods in Section 5 is broad but somewhat shallow. The paper reports aggregate AUROC/AUPRC scores and correctly notes that non-DL methods often perform better. However, it lacks a deep dive into why certain methods fail. A more granular analysis—for example, which specific types of causal links (e.g., highly nonlinear, weak, or lagged) are missed by each algorithm—would provide more actionable insights for future methods development.
2. The authors state that they "performed an initial hyperparameter search" and note that methods requiring less fine-tuning often performed better. This could be a critical confounder. The underperformance of some DL-based methods might stem from suboptimal hyperparameter choices for this new, challenging dataset, rather than a fundamental flaw in the methods themselves. A more systematic approach to hyperparameter optimization on a dedicated validation set would strengthen the claim that simpler methods are inherently superior here.
3. The paper relies on AUROC and AUPRC for graph reconstruction, arguing that other metrics like Structural Hamming Distance (SHD) are difficult to compare across different graph sizes. While this is a reasonable justification, the exclusive reliance on these two metrics may not provide a complete picture of performance. For instance, SHD could still offer valuable information about the absolute number of errors (missing/extra/wrongly oriented edges) within each complexity tier.

**Strengths Contributions:**

1. The paper addresses a critical and well-recognized gap in the causal discovery literature: the lack of a standardized, challenging benchmark for nonlinear, stochastic, and high-dimensional dynamical systems. By providing a robust and extensible framework, CausalDynamics has the potential to significantly accelerate the development and validation of new causal discovery methods, much like ImageNet or CASP did for their respective fields. It provides a crucial bridge between theoretical algorithm development and real-world applicability in science and engineering.
2. The core novelty lies in the benchmark itself and its thoughtful design. The tiered structure (Figure 1) is a major strength, allowing researchers to test algorithms on systems of progressively increasing complexity, from simple ODEs to coupled networks and pseudo-real climate models. The adaptation of the GNR algorithm to generate hierarchically coupled dynamical systems is a novel contribution that allows for the controlled study of specific causal challenges like unobserved confounders and time-lags.
3. The paper is exceptionally well-written, organized, and easy to understand.

---

> ### Author Rebuttal · Authors · 2025-07-30
>
> We thank reviewer Xssv for the valuable comments that contributed to improve this work. In the following we will provide targeted responses to each of the reviewer’s comments.
>
> > The evaluation of SOTA methods in Section 5 is broad but somewhat shallow. The paper reports aggregate AUROC/AUPRC scores and correctly notes that non-DL methods often perform better. However, it lacks a deep dive into why certain methods fail. A more granular analysis—for example, which specific types of causal links (e.g., highly nonlinear, weak, or lagged) are missed by each algorithm—would provide more actionable insights for future methods development.
>
> In the original submission we provided a short comment on how individual methods perform. However, we realise that this is not giving many insights. Thus, we now added:
>
> - Additional text on a more detailed comments model performance to the manuscript
> - A diagnostic tool to extract the full list of metrics to offer a deep dive on method performance for the user
> - Additional experiments: time sampling and partial observability (see additional experiments in rebuttal for reviewer pMkb)
> - 3 more baselines, GRaSP [1], RCD [2], and TCDF [3], as well as SHD [4] as an additional metric
>
> &nbsp;
>
> We now provide more clarification on the **new diagnostics tool**:
>
> We provide the complete metrics for each graph in our HuggingFace data repository, but analyzing each of the 15k+ graphs can be too granular. As a result, we have added a built-in, easy-to-use diagnosis script that allows users to quickly analyze results on the experiment-level. For instance, if users are interested in understanding the effects of unobserved confounders `(confounders=True)` on the performance of different causal discovery algorithms, especially in light of high internal noise/stochasticity `(noise=2.0)`, they can run the following:
>
> ```
> python diagnose.py --exp_dir data/simple/noise=2.00_confounder=True
> ```
> and summary statistics will be logged:
>
> &nbsp;
>
> *Table 1: Diagnostic output*
> | Model     | AUROC | AUPRC | SHD |
> | --------- | ------------ | ------------ | ---------- |
> | cutsplus  | 0.500        | 0.564        | 26.042     |
> | dynotears | 0.491        | 0.592        | 22.750     |
> | fpcmci    | 0.501        | 0.587        | 21.646     |
> | grasp     | 0.507        | 0.574        | 26.083     |
> | ngclstm | 0.499        | 0.564        | 20.271     |
> | pcmciplus | 0.502        | 0.588        | 22.250     |
> | rcd       | 0.500        | 0.564        | 26.042     |
> | tcdf      | 0.500        | 0.564        | 26.042     |
> | tsci      | 0.487        | 0.576        | 26.000     |
> | varlingam | 0.489        | 0.575        | 21.833     |
>
>
> &nbsp;
>
> > The authors state that they "performed an initial hyperparameter search" and note that methods requiring less fine-tuning often performed better. This could be a critical confounder. The underperformance of some DL-based methods might stem from suboptimal hyperparameter choices for this new, challenging dataset, rather than a fundamental flaw in the methods themselves. A more systematic approach to hyperparameter optimization on a dedicated validation set would strengthen the claim that simpler methods are inherently superior here.
>
> This is indeed a crucial point. As such, we **improve on our baseline API wrapper to expose more hyperparameters** allowing us to perform a comprehensive sweep on a held-out validation set. For instance, initializing a DL-based Temporal Causal Discovery Framework (TCDF) model looks something like the below code snippet. Similar improvements are applied across the board on all baseline models, and will likewise be reflected in the documentation.
>
> ```
> class TCDF:
>     def __init__(
>         self,
>         cuda: bool = False,
>         epochs: int = 100,
>         kernel_size: int = 4,
>         hidden_layers: int = 1,
>         learning_rate: float = 0.01,
>         optimizer: str = "Adam",
>         seed: int = 1111,
>         dilation_coefficient: int = 4,
>         significance: float = 0.8,
>         log_interval: int = 500,
>     )
> ```
>
> Finally, **we re-ran all evaluations by first performing hyperparameter search** on a held-out simple dynamics (Tier 1), and using the optimal configuration to score the rest. We find a 10-20% improvement depending on the method. The specific hyperparameter set used for this work is detailed in the Appendix, and reflected as the default initialization values.
>
> &nbsp;
>
> > The paper relies on AUROC and AUPRC for graph reconstruction, arguing that other metrics like Structural Hamming Distance (SHD) are difficult to compare across different graph sizes. While this is a reasonable justification, the exclusive reliance on these two metrics may not provide a complete picture of performance. For instance, SHD could still offer valuable information about the absolute number of errors (missing/extra/wrongly oriented edges) within each complexity tier.
>
> We have **implemented SHD** [4] as an additional metric and re-ran evaluation using AUROC, AUPRC, and SHD scoring criteria to further enrich the interpretation of models’ failure modes. For instance, SHDs for highly noisy and confounded dynamics (Table 1) are higher for deep learning (DL) methods (about 26) while non deep learning methods show lower scores (20-21). Lower scores translate to a better discovery of the true causal links confirming findings from the original submission. This might be because DL-based methods are more sensitive to noisy signals and require more fine-tuning. We thank the reviewer for the metric recommendation.
>
> &nbsp;
>
> > Maybe just out of curious, does this paper: Benchmarking Structural Inference Methods for Interacting Dynamical Systems with Synthetic Data, (accepted by NeurIPS 2024 Dataset and Benchmark track) share similar research interest with this submission?
>
> Yes, the paper is related to our work. However, its focus is on evaluating the robustness and generalizability of causal discovery methods by benchmarking datasets with a limited scope. Our framework goes much further: the hierarchical design allows users to generate custom graphs and trajectories, with the aim of providing a benchmark for evaluating existing methods as well as  for the development of new ones.
>
> We have added this paper as a new reference to our manuscript.
>
> &nbsp;
>
> References:
>
> [1] Wai-Yin Lam, Bryan Andrews, and Joseph Ramsey. *Greedy relaxations of the sparsest permutation algorithm*. In Uncertainty in Artificial Intelligence, 1052–1062. PMLR, 2022.
>
> [2] Takashi Nicholas Maeda, Shohei Shimizu. *RCD: Repetitive causal discovery of linear non-Gaussian acyclic models with latent confounders*. In Proceedings of the Twenty Third International Conference on Artificial Intelligence and Statistics, PMLR 108:735-745, 2020.
>
> [3] Nauta M, Bucur D, Seifert C. *Causal Discovery with Attention-Based Convolutional Neural Networks*. Machine Learning and Knowledge Extraction. 2019; 1(1):312-340.
>
> [4] Ioannis Tsamardinos, Laura E Brown, and Constantin F Aliferis. *The max-min hill-climbing bayesian network structure learning algorithm*. Machine Learning, 65(1):31–78, 2006.

---

> > ### Comment · Reviewer_Xssv · 2025-08-04
> >
> > I would like to thank the authors for addressing my concerns. I believe the new revision will be much better than the original submission.

---

### Official Review · Reviewer_pMkb · 2025-07-01

**Rating:** 4
**Confidence:** 2

**Summary:**

This paper proposes a tiered benchmark and extensible data generation framework for evaluating causal structure discovery in dynamical systems. The benchmark consists of three tiers: Tier 1 includes simple models selected from existing datasets; Tier 2 introduces a Generator-Network-Rewiring (GNR) algorithm to simulate latent confounders and time-lagged interactions; and Tier 3 features pseudo-real climate simulations of coupled atmosphere–ocean dynamics to address high-dimensional challenges. Comprehensive experiments are conducted to evaluate how state-of-the-art causal discovery methods perform across these progressively complex scenarios.

**Dataset Code Accessibility:**

Yes

**Ethical Considerations:**

No, there are no or only very minor ethics concerns

**Final Justification:**

I would like to thank the authors for addressing my concerns. I believe the revised version shows a clear improvement over the original submission, and I will raise my score accordingly.

**Limitations Weaknesses:**

1. The novelty of the benchmark is limited, as it largely builds on existing models, datasets, and public toolkits (e.g., DYSTS, GNR, XRO, QGS); the specific contributions beyond prior resources are not clearly articulated.



2. The selection of baseline methods is insufficient. Several recent advances in causal discovery are omitted, making the empirical comparison less comprehensive.



I appreciate the effort to advance standardized benchmarking for causal discovery in dynamical systems. Clarifying the unique contributions of the proposed benchmark and incorporating a broader set of recent methods would substantially enhance the impact and completeness of the work.

**Strengths Contributions:**

1. The paper provides a well-organized and timely review of related work, positioning the proposed benchmark within the broader landscape of causal discovery in dynamical systems.



2. The tiered design offers a systematic framework to evaluate algorithmic robustness under varying levels of complexity, from simple synthetic models to high-dimensional pseudo-realistic settings.

---

> ### Author Rebuttal · Authors · 2025-07-30
>
> We thank reviewer pMkb for the valuable comments that contributed to improve this work. In the following we will provide targeted comments to address each mentioned limitation.
>
> > The novelty of the benchmark is limited, as it largely builds on existing models, datasets, and public toolkits (e.g., DYSTS, GNR, XRO, QGS); the specific contributions beyond prior resources are not clearly articulated.
>
> We have added additional clarifications in the introduction to further highlight the novelty of this work:
> -  addressing the lack of a standardized, realistic benchmark for specifically nonlinear, stochastic, and high-dimensional dynamical systems.
> - our tiered benchmark design is unique. CausalDynamics is an all-in-one platform to test methods on dynamical systems across a range of complexity (e.g., Lorenz as an idealized representation of complex climate models and Rössler to simulate kinetics in Chemistry).
> - key technical contribution is the adaptation of the GNR algorithm to construct hierarchically coupled dynamical systems, enabling targeted investigation of challenges such as time lags and unobserved confounders.
>
> In summary, our benchmark is in its existence unique and its design is addressing a critical need in causal discovery: the lack of ground truth in real-world systems. To build trust it is essential to evaluate them on fully known synthetic systems that closely resemble real-world dynamics.
>
> >The selection of baseline methods is insufficient. Several recent advances in causal discovery are omitted, making the empirical comparison less comprehensive.
>
> To address this comment we have now extended our benchmark to include **3 additional baseline algorithms**: GRaSP [1], RCD [2], and TCDF [3]. Beyond just adding more baselines, the newly added models represent CausalDynamics’ interoperability with two relatively well-received causal discovery packages: `causal-learn` (GRaSP, RCD) and `temporal causal discovery framework` (TCDF). We ensure that CausalDynamics is fully compatible with their APIs to allow for easy benchmarking and seamless workflow integration. This plays towards our goal of making this benchmark community driven, i.e., enabling the easy implementation of additional methods.
>
> > I appreciate the effort to advance standardized benchmarking for causal discovery in dynamical systems. Clarifying the unique contributions of the proposed benchmark and incorporating a broader set of recent methods would substantially enhance the impact and completeness of the work.
>
> We hope that the above comments have clarified the novelty and unique contributions of this work. In summary:
> - Related recent publications (e.g. [5]) use only a limited number of graphs (11) and ways of interactions (3). CausalDynamics on the other hand offers a more general approach with the option to **create an unlimited number of graphs** with increasing complexity and **flexible connectivity rules**.
> - We have added a **diverse set of SOTA deep-learning and not deep learning-based methods** with seamless integration of other well known causal discovery packages for community-driven extension.
>
> &nbsp;
>
> &nbsp;
>
>
> **Additional experiments**
>
> Following other reviewers’ comments we have further **extended the causal challenges** and **added a third metric** for graph evaluation, SHD [4].
>
> Additional challenges are:
>
> - **time sampling**: We ablate the effect of different time discretization on the performance of causal discovery algorithms. We agree that in many observing systems, the sampling frequency varies even though the measured process has identical causal graph structure for data generation. Testing for an algorithm’s sensitivity to this is another important evaluation that we consider. We, therefore, conduct experiments where we only sample every 1st, 5th, or 10th time step. As an illustration, we consider one of the most difficult experiments: `coupling=nonlinear_noise=2.00_systems=10_confounder=True_standardize=True_timelag=1`.
>
>     &nbsp;
>
>     *Table 1: Ablating time discretization*
>
>     | Model     | AUROC (1 / 5 / 10) | AUPRC (1 / 5 / 10) | SHD (1 / 5 / 10) |
>     | --------- | ------------------------- | ------------------------- | ----------------------- |
>     | pcmciplus | 0.486 / 0.550 / 0.553     | 0.382 / 0.415 / 0.425     | 51.9 / 44.4 / 40.5      |
>     | fpcmci    | 0.547 / 0.557 / 0.566     | 0.406 / 0.418 / 0.429     | 45.65 / 44.2 / 38.9     |
>     | varlingam | 0.519 / 0.518 / 0.548     | 0.389 / 0.389 / 0.407     | 51.85 / 51.25 / 47.2    |
>     | dynotears | 0.546 / 0.539 / 0.574     | 0.426 / 0.416 / 0.444     | 38.9 / 41.6 / 37.85     |
>     | ngclstm | 0.497 / 0.500 / 0.500     | 0.377 / 0.378 / 0.378     | 56.0 / 34.0 / 34.0      |
>     | tsci      | 0.533 / 0.576 / 0.567     | 0.416 / 0.458 / 0.443     | 41.4 / 38.55 / 36.35    |
>     | cutsplus  | 0.495 / 0.500 / 0.500     | 0.376 / 0.378 / 0.378     | 39.0 / 34.0 / 34.0      |
>     | rcd       | 0.500 / 0.501 / 0.505     | 0.378 / 0.382 / 0.385     | 34.0 / 34.1 / 34.05     |
>     | gin       | 0.500 / 0.500 / 0.500     | 0.378 / 0.378 / 0.378     | 34.0 / 34.0 / 34.0      |
>     | grasp     | 0.498 / 0.484 / 0.490     | 0.385 / 0.379 / 0.384     | 35.8 / 36.2 / 36.2      |
>     | tcdf      | 0.500 / 0.500 / 0.500     | 0.378 / 0.378 / 0.378     | 34.0 / 34.0 / 34.0      |
>
>     Overall, we find that reducing sampling frequency tends to improve detection accuracy. This is potentially due to the reduction of data redundancy that can noise the algorithm.
>
>
>     &nbsp;
>
> - **partially observed**: We extend on the existing challenge of hidden confounders. This is a common scenario in real-world situations where our ability to measure the full system is difficult, if not impossible. Specifically, we compare these two cases:
>     - In the partially-observed setting: we sample just one variable per multidimensional node
>     - In the fully-observed setting: we take the mean value of all variables at each node
>
>     &nbsp;
>
>     As an illustration, we consider one of the most difficult experiments: `coupling=nonlinear_noise=2.00_systems=10_confounder=True_standardize=True_timelag=1`.
>
>     &nbsp;
>
>     *Table 2: Ablating partial observability*
>
>     | Model     | AUROC (partial/full) | AUPRC (partial/full) | SHD (partial/full) |
>     | --------- | ------------- | ------------- | --------------- |
>     | pcmciplus | 0.486 / 0.500 | 0.382 / 0.386 | 51.900 / 50.150 |
>     | fpcmci    | 0.545 / 0.546 | 0.405 / 0.406 | 45.800 / 47.350 |
>     | varlingam | 0.519 / 0.503 | 0.389 / 0.381 | 51.850 / 51.650 |
>     | dynotears | 0.546 / 0.559 | 0.426 / 0.426 | 38.900 / 37.200 |
>     | ngclstm | 0.499 / 0.500 | 0.377 / 0.378 | 55.850 / 34.000 |
>     | tsci      | 0.533 / 0.552 | 0.416 / 0.421 | 41.400 / 38.450 |
>     | cutsplus  | 0.496 / 0.500 | 0.376 / 0.378 | 39.150 / 34.000 |
>     | rcd       | 0.500 / 0.499 | 0.378 / 0.377 | 34.000 / 34.000 |
>     | gin       | 0.500 / 0.500 | 0.378 / 0.378 | 34.000 / 34.000 |
>     | grasp     | 0.498 / 0.499 | 0.385 / 0.388 | 35.800 / 36.400 |
>     | tcdf      | 0.500 / 0.500 | 0.378 / 0.378 | 34.000 / 34.000 |
>
>     We find that fully observing the system yields a modest improvement in detection performance. One possible explanation is that, even when some variables are hidden, their influence still propagates through the variables we do observe, so the loss of information is less severe than it might appear.
>
> &nbsp;
>
> References:
>
> [1] Wai-Yin Lam, Bryan Andrews, and Joseph Ramsey. *Greedy relaxations of the sparsest permutation algorithm*. In Uncertainty in Artificial Intelligence, 1052–1062. PMLR, 2022.
>
> [2] Takashi Nicholas Maeda, Shohei Shimizu. *RCD: Repetitive causal discovery of linear non-Gaussian acyclic models with latent confounders*. In Proceedings of the Twenty Third International Conference on Artificial Intelligence and Statistics, PMLR 108:735-745, 2020.
>
> [3] Nauta M, Bucur D, Seifert C. *Causal Discovery with Attention-Based Convolutional Neural Networks*. Machine Learning and Knowledge Extraction. 2019; 1(1):312-340.
>
> [4] Ioannis Tsamardinos, Laura E Brown, and Constantin F Aliferis. *The max-min hill-climbing bayesian network structure learning algorithm*. Machine Learning, 65(1):31–78, 2006.
>
> [5] Wang, A., Tong, T. P., Mizera, A., & Pang, J. (2024). *Benchmarking structural inference methods for interacting dynamical systems with synthetic data*. Advances in Neural Information Processing Systems, 37, 135129-135185.

---

> > ### Comment · Reviewer_pMkb · 2025-08-04
> > **Thank you for author's response**
> >
> > I would like to thank the authors for addressing my concerns. I believe the revised version shows a clear improvement over the original submission, and I will raise my score accordingly.

---

### Official Review · Reviewer_jxcy · 2025-07-02

**Rating:** 5
**Confidence:** 3

**Summary:**

The paper proposes a benchmark for finding causal connectivity structure for dynamical systems. The benchmark consists of a large number of synthetically generated ordinary and stochastic differential equations (ODEs and SDEs) and a few ODE models inspired by climate models. Several methods from the literature were tested on the benchmark.

**Dataset Code Accessibility:**

Yes

**Dataset Code Comments:**

The dataset can readily be downloaded, as well as the accompanying code. In the supplementary material the authors provide a detailed instruction on how to use it.

**Ethical Considerations:**

No, there are no or only very minor ethics concerns

**Final Justification:**

The response of the authors have addressed my concerns about the weakness of the paper/ In particular, their focus on continuous-time models seems justified and they added the case of  systems with partial observations. They also addressed my concerns regarding their focus on chaotic systems.

**Limitations Weaknesses:**

The first limitation of the benchmark is the focus on continuous-time models. However, a large part of the literature focuses on discrete-time models. At the same time, it is known that time discretisation may destroy the original adjacency matrix:

Roebroeck, A. F., Formisano, E., & Goebel, R. W. (2011). Reply to Friston and David After comments on:
The identification of interacting networks in the brain using fMRI: Model selection, causality and
deconvolution. Neuroimage, 58(2), 310-311. https://doi.org/10.1016/j.neuroimage.2009.10.07

For this reason, including discrete-time models with known ground truth would also have been useful. Another weakness is that the benchmark models all assume observation of all the states. While observing only certain states can probably be easily added, it would have been nice to have examples of partially observed dynamical systems with the same input-output behavior but different connectivity graph. In addition, if I understood it correctly, the benchmarks are chaotic. I wonder if it is really a good benchmark, as in general learning chaotic dynamical systems in a reliable manner  is problematic, especially in the presence of hidden states, and as a result most of the literature considers stable dynamical systems.

**Strengths Contributions:**

As the authors have pointed out, the field lacks benchmarks with a known ground truth and with a large  and number variety of models. The proposed benchmark fills the gap.

---

> ### Author Rebuttal · Authors · 2025-07-30
>
> We thank reviewer jxcy for the valuable comments that contributed to improve this work. In the following we will provide targeted comments to address each mentioned limitation.
>
> > The first limitation of the benchmark is the focus on continuous-time models. However, a large part of the literature focuses on discrete-time models. For this reason, including discrete-time models with known ground truth would also have been useful.
> Roebroeck, A. F., Formisano, E., & Goebel, R. W. (2011). Reply to Friston and David After comments on: The identification of interacting networks in the brain using fMRI: Model selection, causality and deconvolution. Neuroimage, 58(2), 310-311. https://doi.org/10.1016/j.neuroimage.2009.10.07.
>
> Indeed, discrete-time models are prevalent in literature and occur widely across fields (e.g., tabular data of non-continuous processes in socioeconomic, political surveys, randomized controlled trials in healthcare). However, at this point, we want to position our work as a complement to this more established field, particularly to advance the field of causal discovery for continuous time-series data that poses its unique idiosyncrasies (e.g., autocorrelation, non-independent noise structures, evolving causal graphs).
>
> > At the same time, it is known that time discretization may destroy the original adjacency matrix.
>
> To address reviewers jxcy’s comments on **time discretization**, we have added an additional challenge to the benchmark. In particular, we ablate the effect of different time sampling on the performance of causal discovery algorithms. We agree that in many observing systems, the sampling frequency varies even though the measured process has identical data-generating causal graphs. Testing for an algorithm’s sensitivity to this is another important evaluation that we consider. We, therefore, conduct experiments where we only sample every 1st, 5th, or 10th time step. As an illustration, we consider one of the most difficult experiments: `coupling=nonlinear_noise=2.00_systems=10_confounder=True_standardize=True_timelag=1`. Please note that following other reviewers’ comments we have also **added 3 more baseline algorithms**, including GRaSP [1], RCD [2], and TCDF [3], as well as SHD [4] as an **additional metric**.
>
> Table 1: Ablating time discretisation
> | Model     | AUROC (1 / 5 / 10) | AUPRC (1 / 5 / 10) | SHD (1 / 5 / 10) |
> | --------- | ------------------------- | ------------------------- | ----------------------- |
> | pcmciplus | 0.486 / 0.550 / 0.553     | 0.382 / 0.415 / 0.425     | 51.9 / 44.4 / 40.5      |
> | fpcmci    | 0.547 / 0.557 / 0.566     | 0.406 / 0.418 / 0.429     | 45.65 / 44.2 / 38.9     |
> | varlingam | 0.519 / 0.518 / 0.548     | 0.389 / 0.389 / 0.407     | 51.85 / 51.25 / 47.2    |
> | dynotears | 0.546 / 0.539 / 0.574     | 0.426 / 0.416 / 0.444     | 38.9 / 41.6 / 37.85     |
> | ngclstm | 0.497 / 0.500 / 0.500     | 0.377 / 0.378 / 0.378     | 56.0 / 34.0 / 34.0      |
> | tsci      | 0.533 / 0.576 / 0.567     | 0.416 / 0.458 / 0.443     | 41.4 / 38.55 / 36.35    |
> | cutsplus  | 0.495 / 0.500 / 0.500     | 0.376 / 0.378 / 0.378     | 39.0 / 34.0 / 34.0      |
> | rcd       | 0.500 / 0.501 / 0.505     | 0.378 / 0.382 / 0.385     | 34.0 / 34.1 / 34.05     |
> | gin       | 0.500 / 0.500 / 0.500     | 0.378 / 0.378 / 0.378     | 34.0 / 34.0 / 34.0      |
> | grasp     | 0.498 / 0.484 / 0.490     | 0.385 / 0.379 / 0.384     | 35.8 / 36.2 / 36.2      |
> | tcdf      | 0.500 / 0.500 / 0.500     | 0.378 / 0.378 / 0.378     | 34.0 / 34.0 / 34.0      |
>
> Overall, we find that reducing sampling frequency tends to improve detection accuracy. This is potentially due to the reduction of data redundancy that can noise the algorithm.
>
> > Another weakness is that the benchmark models all assume observation of all the states. While observing only certain states can probably be easily added, it would have been nice to have examples of partially observed dynamical systems with the same input-output behavior but different connectivity graph.
>
> In our benchmark, we have already included a hierarchy of **hidden confounding** challenges, such as unobserved forcings (periodic or linear), or partial observations (only observing one variable in the multidimensional node). However, to explicitly address the comment of a changing graph structure but maintaining the input-output relationship, we have extended on the challenge to a direct comparison of partial and full observations:
>
> - In the partially-observed setting: we sample just one variable per multidimensional node
>
> - In the fully-observed setting: we take the mean value of all variables at each node
>
> As an illustration, we consider one of the most difficult experiments: `coupling=nonlinear_noise=2.00_systems=10_confounder=True_standardize=True_timelag=1`.We find that fully observing the system yields a modest improvement in detection performance. One possible explanation is that, even when some variables are hidden, their influence still propagates through the variables we do observe, so the loss of information is less severe than it might appear.
>
> Table 2: Ablating partial observability
> | Model     | AUROC (partial/full) | AUPRC (partial/full) | SHD (partial/full) |
> | --------- | ------------- | ------------- | --------------- |
> | pcmciplus | 0.486 / 0.500 | 0.382 / 0.386 | 51.900 / 50.150 |
> | fpcmci    | 0.545 / 0.546 | 0.405 / 0.406 | 45.800 / 47.350 |
> | varlingam | 0.519 / 0.503 | 0.389 / 0.381 | 51.850 / 51.650 |
> | dynotears | 0.546 / 0.559 | 0.426 / 0.426 | 38.900 / 37.200 |
> | ngclstm | 0.499 / 0.500 | 0.377 / 0.378 | 55.850 / 34.000 |
> | tsci      | 0.533 / 0.552 | 0.416 / 0.421 | 41.400 / 38.450 |
> | cutsplus  | 0.496 / 0.500 | 0.376 / 0.378 | 39.150 / 34.000 |
> | rcd       | 0.500 / 0.499 | 0.378 / 0.377 | 34.000 / 34.000 |
> | gin       | 0.500 / 0.500 | 0.378 / 0.378 | 34.000 / 34.000 |
> | grasp     | 0.498 / 0.499 | 0.385 / 0.388 | 35.800 / 36.400 |
> | tcdf      | 0.500 / 0.500 | 0.378 / 0.378 | 34.000 / 34.000 |
>
> > In addition, if I understood it correctly, the benchmarks are chaotic. I wonder if it is really a good benchmark, as in general learning chaotic dynamical systems in a reliable manner is problematic, especially in the presence of hidden states, and as a result most of the literature considers stable dynamical systems.
>
> We agree chaotic systems pose a challenge to causal discovery methods. Most real-world physical systems are inherently chaotic rather than stable. A reliable causal discovery method should be robust to changes in initial conditions or internal perturbations, as long as the underlying causal structure remains identical, especially since real-world observations are rarely stable (e.g., climate change due to external anthropogenic forcing). Finally, if a user is interested in an equilibrium system, one can simply discard the initial segment of the time series (e.g., the first 500 steps) to allow the system to evolve along any stable attractors.
>
> We want to highlight that the task is not to predict exact trajectories of chaotic systems, which would indeed be impossible beyond the deterministic time horizon, but rather to uncover structural causal connections, which is challenging but manageable. The fact that it is a challenge is exactly the reason why chaotic systems are a good foundation for a benchmark. It forces the community to advance their methods similar to the ARC challenge, which is extremely difficult for ML models right now, but pushes the frontier of research.
>
> References:
>
> [1] Wai-Yin Lam, Bryan Andrews, and Joseph Ramsey. *Greedy relaxations of the sparsest permutation algorithm*. In Uncertainty in Artificial Intelligence, 1052–1062. PMLR, 2022.
>
> [2] Takashi Nicholas Maeda, Shohei Shimizu. *RCD: Repetitive causal discovery of linear non-Gaussian acyclic models with latent confounders*. In Proceedings of the Twenty Third International Conference on Artificial Intelligence and Statistics, PMLR 108:735-745, 2020.
>
> [3] Nauta M, Bucur D, Seifert C. *Causal Discovery with Attention-Based Convolutional Neural Networks*. Machine Learning and Knowledge Extraction. 2019; 1(1):312-340.
>
> [4] Ioannis Tsamardinos, Laura E Brown, and Constantin F Aliferis. *The max-min hill-climbing bayesian network structure learning algorithm*. Machine Learning, 65(1):31–78, 2006.

---

> > ### Comment · Reviewer_jxcy · 2025-08-07
> >
> > I thank the authors for addressing my comments regarding the weaknesses of the paper. I adjust my score accordingly

---

### Note · Authors · 2025-08-13

We would like to thank the reviewers for taking the time to review and consider our rebuttals. They do greatly improve our work!
In our revision, we incorporated the feedback which led to the addition of:

- More challenges: time subsampling, partial observability experiments
- More baseline models that are interoperable with existing causal discovery packages
- More evaluation metric, e.g., SHD
- An easy-to-use tool to investigate granular, graph-level results and possible limitations
- An expanded API for more comprehensive hyperparameter search and generating new experimental results
- Addition of nonlinearities by implementing various edge-level activations


In particular, we are grateful for and encouraged by the positive comments in that CausalDynamics *“fills existing gap”* (Reviewer jcxy), *“offers a systematic framework to evaluate algorithmic robustness under varying levels of complexity”* (Reviewer pMkb), has *“thoughtful design”*, is *“exceptionally well-written”* (Reviewer Xssv), and *“clearly understandable”* (Reviewer Vt55).


We hope that our work will keep on continuously improving through community feedback in the future to push the boundaries of robust causal discovery methods for real world scenarios.

---

### Decision · Program_Chairs · 2025-09-18

**Decision:**

Accept (poster)

**Comment:**

There is a consensus among all reviewers to accept this paper, which I agree.

===== FINAL UPDATE FROM DB Track PCs ====

The final decision for this paper has been taken by the program chairs after consultation with the SACs. All Senior Area Chairs have ranked papers according to the feedback from the AC during the review process. We decided to leave the original meta-review to reflect the opinion of the AC in light of the initial discussions with reviewers and SAC.